# Deep-sea gas hydrate mounds and chemosynthetic fauna discovered at 3640 m on the Molloy Ridge, Greenland Sea

Giuliana Panieri [1,21] ✉, Jonathan T. Copley [2] ✉, Katrin Linse[3], Verity Nye[4], Eva Ramirez-Llodra[5], Claudio Argentino [1], Bénédicte Ferré[1], the Arctic Deep - Extreme24 consortium* & Alex D. Rogers[4,22]

Methane seepage at the seafloor can form gas hydrate and sustain chemosynthetic communities of deep-sea animals. Most known hydrate seeps occur shallower than 2000 m on continental slopes, whereas hydrothermal vents are found at greater depths along active spreading centres. Here we report the discovery of hydrate mounds with cold-seep fauna at 3640 m deep on the Molloy Ridge. The mounds display seafloor morphologies resulting from progressive stages of hydrate dissociation. Gas bubbles from the mounds rise to within 300 m of the ocean surface, and isotopic analysis shows the hydrates contain thermogenic gas. Crude oil sampled from the hydrate deposits indicates a young Miocene source rock formed in a fresh-brackish water paleo-environment. The hydrate mounds are inhabited by taxa including siboglinid and maldanid tubeworms, skeneid and rissoid snails, and melitid amphipods. Family-level composition of the fauna is similar to that of Arctic hydrothermal vents at similar depths, including the Jøtul vent field on the Knipovich Ridge, and less similar to nearby methane seeps at shallower depths. The overlap between seep and vent fauna in the Arctic has implications for understanding ecological connectivity across deep-sea habitats and assessing their vulnerability to future impacts from seafloor resource extraction in the region.

Gas hydrates are crystalline solids formed from water and gas molecules under high-pressure and low-temperature conditions[1]. They are abundant in marine sediments along continental margins, typically occurring at water depths greater than 400 m, but in the Arctic, they can remain stable on the seafloor at depths as shallow as ~300 m because of the low bottom-water temperatures[2]. While there is theoretically no maximum depth limit for the stability of seafloor hydrates because of increasing pressure and consistently low bottom-water temperatures, most discovered outcrops occur at depths shallower

than 2000 m on continental slopes, where rapid burial of organic matter leads to the formation of hydrocarbon reservoirs. These hydrocarbon accumulations migrate through faults or low-permeability sedimentary pathways towards the seafloor, feeding gas hydrate systems[2,3]. Gas hydrates constitute an essential global carbon reservoir, estimated to contain $(1–5) \times 10^{15} \, m^3$ or ~500–2500 Gt $(10^{15} \, g)$ C, and are a potential source of atmospheric methane, a potent greenhouse gas[4]. The gas within hydrates can derive from biodegradation of sedimentary organic matter, such as in deposits on the

[1]Department of Geosciences, UiT The Arctic University of Norway, Tromsø, Norway. [2]School of Ocean & Earth Science, University of Southampton, Southampton, UK. [3]British Antarctic Survey, Cambridge, UK. [4]Ocean Census, Begbroke Science Park, Oxfordshire, UK. [5]REV Ocean, Fornebu, Norway. [21]Present address: CNR, ISP Institute of Polar Science, Campus Scientifico-Università Ca' Foscari Venezia, Mestre, Italy. [22]Present address: National Oceanography Centre, Southampton, UK.*A list of authors and their affiliations appears at the end of the paper. ✉e-mail: giuliana.panieri@uit.no; jtc@southampton.ac.uk

Blake Ridge (NW Atlantic) and Cascadia margin (NE Pacific), or be thermogenic in origin, formed by decomposition of organic molecules under high temperature and pressure in deep sedimentary strata[5], as found in deep deposits in the Gulf of Mexico[4].

Gas hydrate systems are associated with cold seeps, where biogeochemical processes support locally abundant populations of specialised fauna that rely on in situ prokaryotic chemosynthetic primary production[6]. These chemosynthetic communities are typically dominated by tubeworms, bivalves and gastropods, in association with bacteria capable of metabolising methane, sulphide produced by anaerobic oxidation of methane and higher hydrocarbons coupled with sulphate reduction, and other hydrocarbons[7]. Seep communities influence local biodiversity, particularly in the relatively species-poor Arctic deep sea[8].

Cold-seep communities in the Arctic have been described from the Håkon Mosby Mud Volcano on the western margin of the Barents Sea at 72.0 °N and 1250 m depth[9], and from methane seeps associated with subsurface hydrates at Vestnesa Ridge on the continental slope of the Fram Strait at 79.1 °N and 1200 m depth[10,11]. Methane seepage also occurs on the Svyatogor Ridge, a sediment-covered transform fault on the flanks of the Knipovich Ridge at 79.4 °N and ~1900 m depth. Svyagotor Ridge hosts the deepest cold-seep community found in the Arctic so far[12], although its fauna has not been characterised in detail at the time of our analyses[13]. In shallower waters, exposed hydrate mounds occur at the Storfjordrenna site on the western margin of the Barents Sea at a depth of 350–390 m[14] and other methane seeps and mud volcanoes are present from 70 to 800 m depth in areas including the Barents Sea[15,16], Beaufort Sea, and canyons on the Norwegian continental margin[8]. The fauna of these Arctic cold seeps includes siboglinid tubeworms, thyasirid clams, and rissoid snails[8,17], and the seeps at shallower depths are often inhabited by abundant populations of species known from non-chemosynthetic habitats[8].

Six active deep-sea (>200 m depth) hydrothermal vent fields are currently confirmed above latitude 70 °N. The Soria Moria (500–550 m depth) and Troll Wall (700–750 m depth) sites are 5 km apart at 71 °N on the southern end of Mohns Ridge and are occupied largely by taxa known from non-chemosynthetic habitats[18]. Vent fauna has not yet been characterised at the Aegir vent field at depth 2600 m and 72.3 °N on Mohns Ridge[19], nor at the Jøtul vent field at depth 3020 m and 77.4 N on the Knipovich Ridge[20]. The fauna at Loki's Castle at depth 2350 m and 73.6 °N on the northern end of Mohns Ridge[21], and the Aurora Vent Field at depth 3888 m and 82.9 °N on the Gakkel Ridge[22], includes some taxa not previously recorded at nearby seeps, such as melitid amphipods and cocculinid limpets[22–24].

Previously, it has been concluded that vents and seeps share relatively few species, although similarities in faunal composition at the level of genera and families suggests evolutionary links such as common ancestry with slope fauna or dispersal from one chemosynthetic system to another[25,26] However, this view was likely influenced by the lack of sampling and, more recently, where vents and seeps occur in close proximity and at similar depths higher faunal similarities have been observed[27,28]. Shared taxa seem especially likely to occur where similar habitats occur on vents and seeps, with sedimented vent sites appearing to share a particularly high number of taxa with those at seeps[28]. The proximity of lower bathyal vents and seeps in the Arctic raises the possibility of closer connectivity between these ecosystems north of latitude 73°N, depending on whether they share habitat characteristics.

The Molloy Ridge is a slow to ultraslow spreading centre in the Fram Strait, extending north for ~60 km from the Molloy Fracture Zone at ~79.1 °N to the Spitsbergen Fracture Zone at ~79.7 °N[29]. The seafloor depth of the ridge axis varies from ~5000 m at its southern end, rising to ~1500 m on an Oceanic Core Complex midway along the ridge, and descending to ~4000 m at the northern end[29]. The formation of the Molloy Ridge began after the opening of the Norwegian-Greenland Sea

at ~56 Mya[30], and most likely the seafloor spreading at the current Molloy Ridge started at ~20 Mya[31].

At the northern end of the Molloy Ridge and in the Spitsbergen Fracture Zone, two large plumes of gas bubbles, described as gas flares, have been detected acoustically, rising ~1770 and ~3355 m above the seafloor, respectively, with the larger plume representing the tallest known worldwide[32]. From seismic reflection data, these plumes were hypothesised to consist of bubbles of oil-associated thermogenic gas[32]. The seafloor sources of the plumes, which occur at >3000 m depth, have not been characterised yet.

During the *Ocean Census Arctic Deep – EXTREME24* expedition in May 2024, we investigated the seafloor source of the water column gas flare using shipboard instruments and a deep-diving Remotely Operated Vehicle (ROV). We discovered exposed hydrate mounds, named the Freya gas hydrate mounds, inhabited by chemosynthetic fauna at a depth of 3640 m (Fig. 1). These represent the deepest known hydrate deposits worldwide. Methane seepage and crude oil were directly observed and sampled with the ROV, revealing hydrocarbon seepage supporting chemosynthetic life ~1770 m deeper than any other Arctic cold seeps and at depths comparable with the nearby high-Arctic hydrothermal vents in the region[20–22].

Here we present the results from geochemical analysis of hydrates and oil collected from the Freya mounds, which clarifies the origins of the hydrocarbons being released from this site into the overlying ocean. Based on seafloor observations, we identify a sequence of morphological evolution of these hydrate features from inception to collapse. We also characterise the fauna colonising this deep methane seep and compare its taxonomic composition with chemosynthetic communities at other Arctic cold seeps and hydrothermal vents, including the first faunal samples collected from the nearby Jøtul vent field as part of this study. Our results provide insights into the geology and ecology of these habitats and their regional context for understanding patterns of deep-sea biodiversity in the Arctic.

## Results
### Discovery of the Freya gas hydrate mounds
Shipboard multibeam echosounder (MBES) data confirmed the presence of gas flares above the Molloy Ridge at 79.6930 °N 3.6617°E (Fig. 1a, b), which were originally detected by the Norwegian MAREANO programme[33]. This location corresponds with the gas flare designated 'GFA' in ref. 32. Multibeam backscatter detected two bubble plumes reaching a minimum depth of ~290 m, where the water temperature recorded in the CTD profile was 2.63 °C (Fig. 2).

An ROV survey targeting the seafloor beneath the flares revealed the presence of three gas hydrate mounds, two pit-like collapse features and a few small ridges within an area of ~100 × 100 m at depth from 3570 to 3747 m (Fig. 1b). The ROV's sonar and visual observations confirmed gas seepage adjacent to the mounds (Fig. 2 and Supplementary Movie 1), thus linking the water column observations to the seafloor hydrates. The temperature measured at 12 m altitude above the seafloor in a CTD profile was −0.63 °C (Fig. 2).

### Mound morphology and hydrocarbon (oil and gas) composition
The mounds investigated are conical in shape, ~4–6 m in diameter and ~2–4 m high (Fig. 3a, b). They are covered by a thin layer of soft sediment, occasionally by carbonate slabs, and colonised by siboglinid and maldanid tubeworms that appear to stabilise the surface. Morphological variations of the mounds indicate a development sequence from sedimented domes with no exposed hydrate (Fig. 3a) to mounds with exposed hydrate in the summit (Fig. 3b) and more eroded or decomposed mounds resulting in arches and cave-like structures (Fig. 3c). We also noticed pit-like collapse features ~6–8 m in diameter (Fig. 3e) and several small ridges, rising just a few decimetres off the seafloor and spanning approximately 1–2 m.

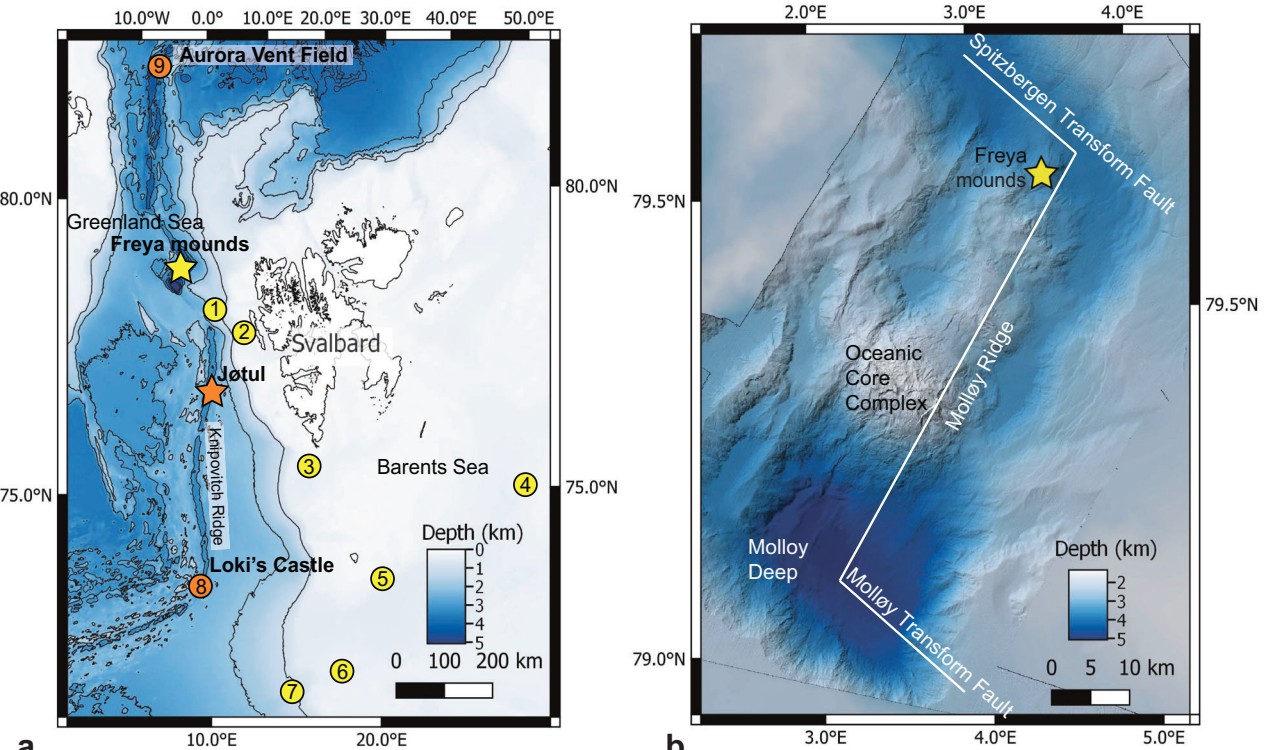

**Fig. 1 | Overview of the location of high-Arctic (>72 °N) cold seeps and hydrothermal vents. a** regional map of seeps (yellow) and vents (orange): yellow star = Freya gas hydrate mounds; orange star = Jøtul vent field 1 = Vestnesa Ridge seeps; 2 = Prins Karls Forland seeps; 3 = Storfjordrenna gas hydrate mounds; 4 = Bjørnøyrenna seeps; 5 = Leirdjupet Fault Complex seeps; 6 = Borealis Mud

Volcano; 7 = Håkon Mosby Mud Volcano; 8 = Loki's Castle; 9 = Aurora Vent Field. Seabed topography shown is from the Global Multi-Resolution Topography (GMRT) synthesis[70]. **b** map of seafloor features observed during ROV dives at the Freya gas hydrate mounds (79.6 °N, depth 3640 m). Detailed bathymetry from MAREANO/Norwegian Mapping Authority[71].

The hydrate structure hosts visibly trapped gas bubbles. In some portions, the hydrate is yellow and white in white-balanced ROV video images[1] (Fig. 3c). Aboard the research ship, as soon as we opened the blade corer, we observed the decomposition of the hydrate, which had already started during the ascent of the ROV, and persisted for several minutes while we were collecting the gas hydrate samples. We collected four hydrate subsamples containing methane ($C_1$, ~66%), accompanied by smaller amounts of ethane ($C_2$, ~8%), propane ($C_3$, ~14%), isobutane (i-$C_4$, ~3%), and normal butane ($C_4$, ~2.3%) yielding an average $C_1/(C_2 + C_3)$ ratio of 3.0 (Fig. 4). The isotopic composition of the gas confirmed an oil-associated thermogenic origin resulting in methane with $\delta^{13}C$ of −47‰ ($n = 4$; 1s = 0.8‰) and $\delta D$ of −188.5‰ (1s = 1.7‰) and heavy $\delta^{13}C$ composition of $CO_2$ of 0.6‰ (1s = 0.2‰) (Fig. 4). The oil present in the hydrate samples shows a characteristic alkane distribution associated with gas condensate, with alkane chain lengths $C_{13}$ (Supplementary Fig. 2). Steranes and diasteranes indicate a source rock deposited in a fresh/brackish lacustrine environment (tetracyclic polyprenoids-TPP and $C_{26}/C_{25}$ tricyclic terpanes ratios) with minor marine contribution (24-n-propylcholestane and 4-methylsteroids relatively sparse) (Supplementary Fig. 3). Moreover, the abundant oleanane and ursane compounds are consistent with high angiosperm inputs, with only traces of gymnosperm diterpanes, suggesting to a Miocene or younger source. One oil-impregnated sediment sample collected to study fauna displayed a more open marine organic signature with the presence of immature higher plants (Supplementary Figs. 2 and 3). The maturity proxies indicated a wet gas/pre-oil maturity window (Supplementary Fig. 4).

## Biological community composition

More than 20 faunal morphospecies were observed at the methane hydrate site, as detailed in Table 1. The upper surfaces and periphery of

the hydrate mounds are conspicuously colonised by dense aggregations of the sessile siboglinid polychaete *Sclerolinum* cf. *contortum* (Fig. 5a), termed the '*Sclerolinum* forest', a refinement of the 'tube-worm forest' concept introduced by[34], and more dispersed maldanid polychaetes (Fig. 5b) in soft sediments.

Invertebrates, including melitid amphipods (Fig. 5c), caridean shrimps, pycnogonids, and nemertean worms, were found in association with the *Sclerolinum* forest and maldanid polychaete tubes. Other polychaetes sampled from sediments at the mounds include an ampharetid species (Fig. 5d). Stauromedusae (Fig. 5e), identified as *Lucernaria* cf. *bathyphila*, were observed within the *Sclerolinum* forest on the methane hydrate mounds and among the maldanid polychaete tubes. Smaller specimens of the stauromedusa were also found in samples of the *Sclerolinum* forest collected by the ROV.

High densities of rissoid and skeneid microgastropods, each 2–3 mm in size, were noted in samples of the *Sclerolinum* forests and attached to maldanid tubes (Fig. 5f). The shell of the rissoid gastropod morphospecies was coated in orange precipitate, whereas the skeneid gastropod morphospecies featured a hyaline shell revealing light-coloured soft parts and white gonadal tissue at its apex. The same habitats also commonly hosted a buccinid gastropod, with juveniles smaller than 1 mm found in the *Sclerolinum* forests and larger specimens observed on *Sclerolinum* and maldanid tubes.

Dead thyasirid bivalves were observed on the sediment surface at the mounds, while live specimens (Fig. 5g) were retrieved using ROV push cores and scoops next to the *Sclerolinum* forest. Additionally, a smaller bivalve species with a maximum shell size of 1.5 mm and a black precipitate coating was found in the same area. Other taxa observed at the Freya mounds include the stalked sponge *Caulophacus* cf. *arcticus* and the fishes *Lycodes* cf. *frigidus* and *Lycenchelys* cf. *platyrhina*.

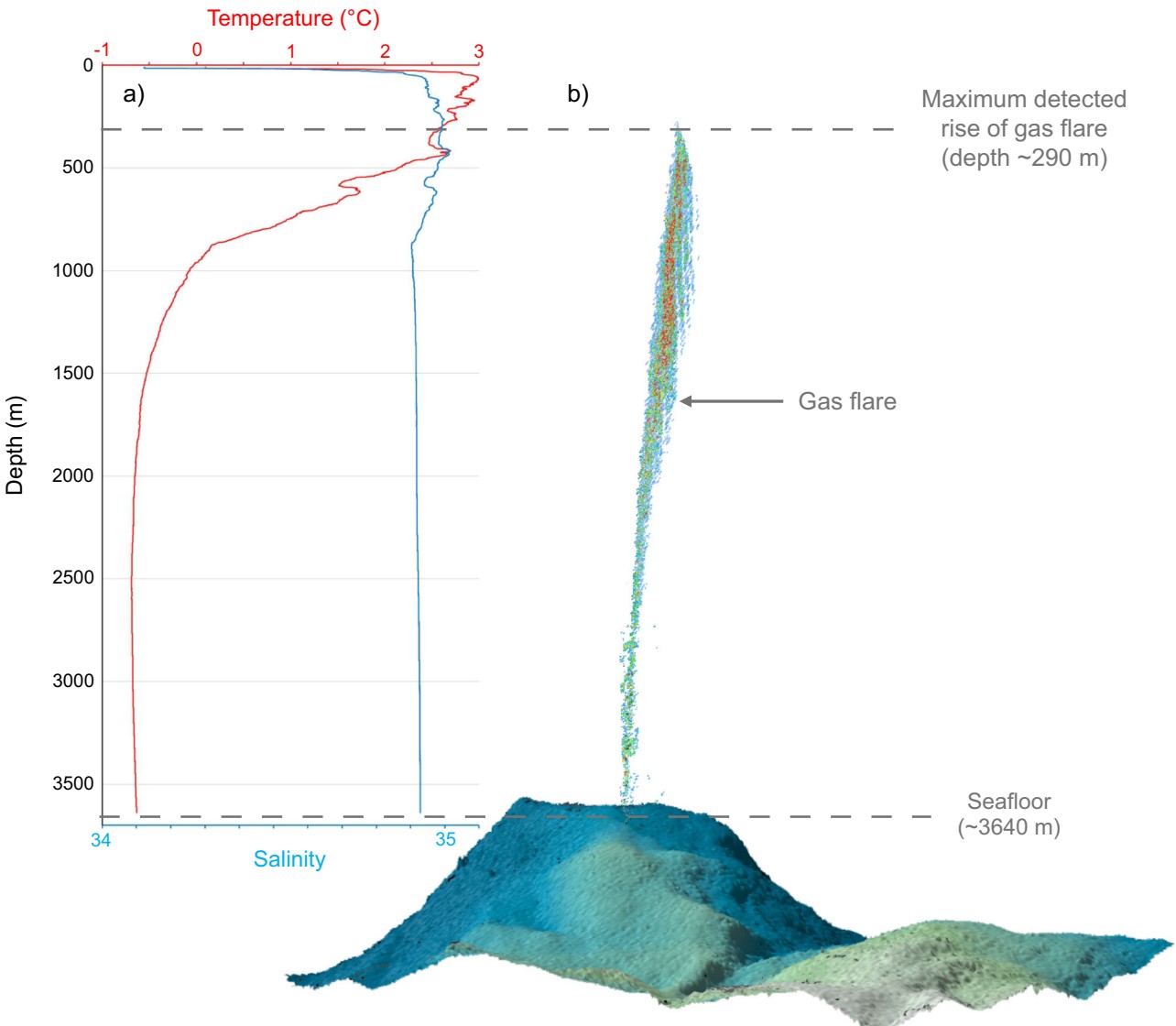

**Fig. 2 | Water column characteristics at the Freya gas hydrate mounds. a** Depth profiles of temperature (red) and salinity (blue) measured by CTD, and **b** is topography as processed with Qimera and acoustic backscatter processed with FMMidwater using the shipboard multibeam echosounder (MBES) at the Freya gas hydrate mounds (79.6 °N, depth 3640 m). The dashed lines show the seafloor depth and the maximum depth of the top of the flare.

## Comparison of Freya fauna with other Arctic seeps and vents

In addition to the discovery and investigation of the Freya gas hydrate mounds, our expedition described the fauna from the Jøtul hydrothermal vent field. This vent field is situated 266 km south of the Freya mounds at a depth of 3020 m on the Knipovich Ridge (Fig. 1a). The fauna at the Jøtul vents includes the siboglinid tubeworm *Sclerolinum* cf. *contortum* with melitid amphipods, caridean shrimp, skeneid and rissoid snails, which are also present in the fauna at the Freya mounds (Supplementary Data 1). At the family level, the fauna that we sampled at the Jøtul hydrothermal vents shows a 59% Sørensen Index similarity with the Freya mound fauna (Fig. 6).

In comparison with faunal inventories compiled for other seeps and vents from published literature (Supplementary Data 1), the fauna identified to family level at Freya and Jøtul is most similar to the fauna recorded at Loki's Castle vent field (47% single-linkage Sørensen Index similarity) and Vestnesa Ridge seeps (46% single-linkage Sørensen Index similarity), and least similar to the fauna at the Prins Karls Forland (PKF) seeps (23% single-linkage Sørensen Index similarity; Fig. 6). Several widespread taxa contribute to faunal similarity between sites,

including habitat-engineering tubeworms (Siboglinidae: recorded at all sites except the Aurora Vent Field; and Maldanidae: recorded at five out of eight sites) and rissoid snails (recorded at six out of eight sites; Supplementary Data 1).

The proximity of sites, calculated as great-circle distances from latitude and longitude values, does not correlate significantly with Sørensen Index similarities (Spearman rank correlation: $r_s = -0.24$, $p = 0.21$, 26 d.f.). However, differences in depth between sites show a significant negative correlation with faunal similarity values (Spearman rank correlation: $r_s = -0.47$, $p = 0.012$, 26 d.f.), indicating that depth may be a factor influencing faunal composition.

The number of families recorded at sites varies from 5 at the Aurora Vent Field to 46 at the Håkon Mosby Mud Volcano (Supplementary Data 1), which may result from greater cumulative sampling effort at longer-studied sites. But there is no significant negative correlation between faunal similarity and differences in family richness between sites (Spearman rank correlation: $r_s = -0.089$, $p = 0.65$, 26 d.f.), indicating that variation in the number of families recorded at sites does not determine their overall pattern of faunal similarity.

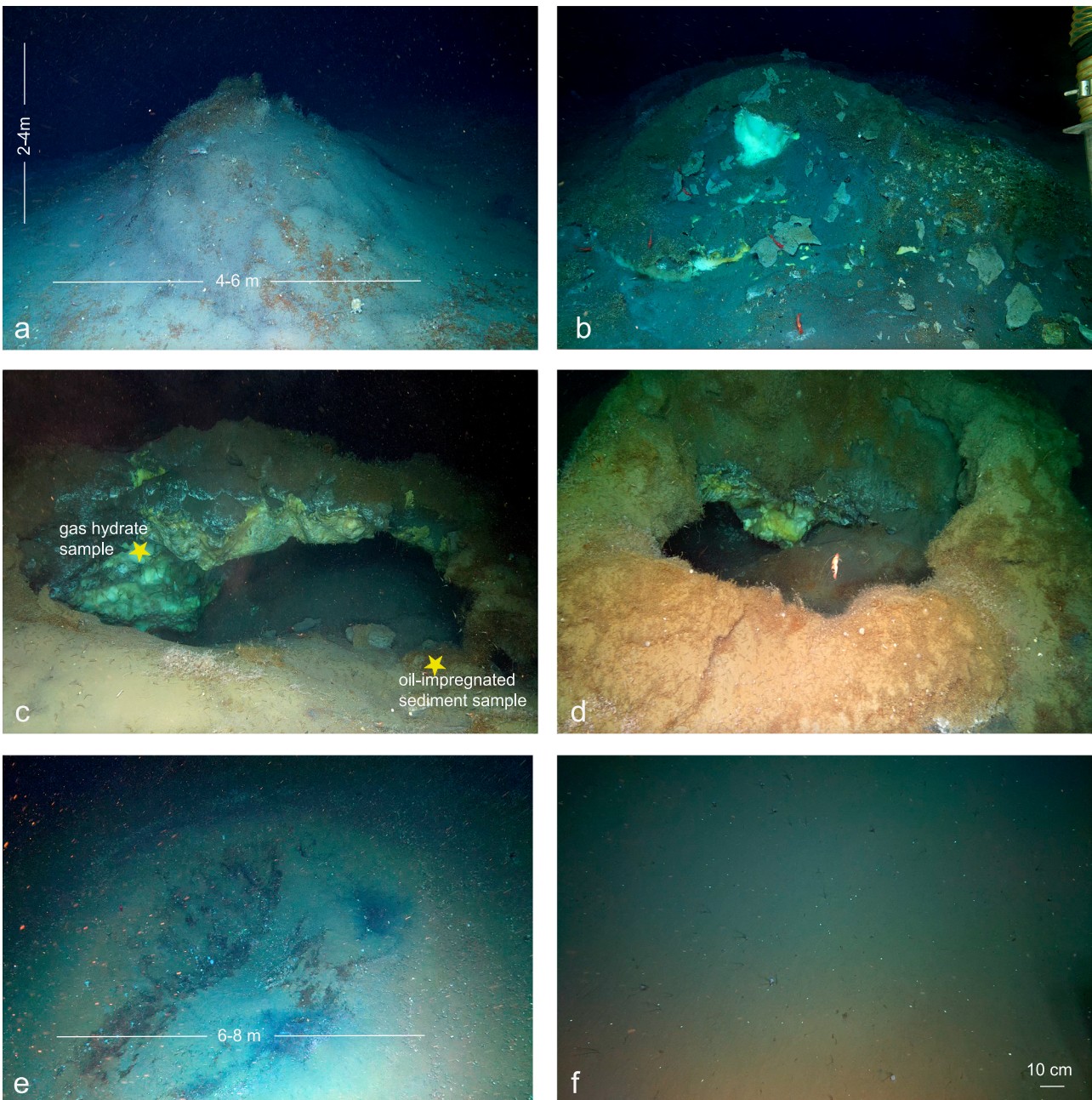

**Fig. 3 | Freya gas hydrate mounds showing different morphologies.** The mounds, made of hydrates, are covered by sediments and frenulate tubeworms forming a '*Sclerolinum* forest' (**a**) with occasionally amphipods and caridean red shrimp (**b**, **d**). Sometimes, around and at the top of the mounds, there are centimetric carbonate crusts (**b**). **c** Shows the position where the sample of gas hydrate for geochemical analyses was taken (yellow star; Supplementary Fig. 1) and the sediment sample used for faunal identification, that on board also revealed the presence of oil. **c**, **d** The influence of hydrate buoyancy on mound morphology that leads to structural fractures and alterations in the integrity of the mounds, ultimately resulting in the formation of collapse-like features (**e**). **f** Background seafloor.

## Discussion

The direct evidence of hydrate outcrops at unprecedented depths producing gas flares that rise for more than 3000 m to within 300 m of the ocean surface, confirms the active nature of these features and their potential contribution to carbon cycling in the water column. The presence of another gas flare nearby in the Spitsbergen Transform Fault[35] also indicates a likelihood of further methane seep communities at >3000 m depth in the region, possibly associated with gas hydrates. Studying these Arctic ultra-deep gas hydrate systems is crucial to enhance our understanding of the deep carbon cycling and ecosystems influenced by natural hydrocarbon emissions, which is key to fill gaps in Arctic deep-sea biogeography.

## Composition and dynamics of the Freya gas hydrate mounds

The Freya gas hydrate mounds contain thermogenic gas primarily composed of methane ($C_1$) and a smaller amount of heavier hydrocarbons ($C_2$–$C_5$). This thermogenic gas is produced from the degradation of organic matter under high heat and pressure conditions and migrates upward through faults in the area, as indicated by previous studies[32], acting as conduits from deeper geological strata to shallower sediment layers where gas hydrates form. Geochemical analysis indicates that the oil, and possibly the associated gas, originated from the breakdown of material derived from angiosperms, flowering plants that were abundant in the Arctic during the Miocene epoch[36]. We draw a first-order correlation with the potential source rock identified for

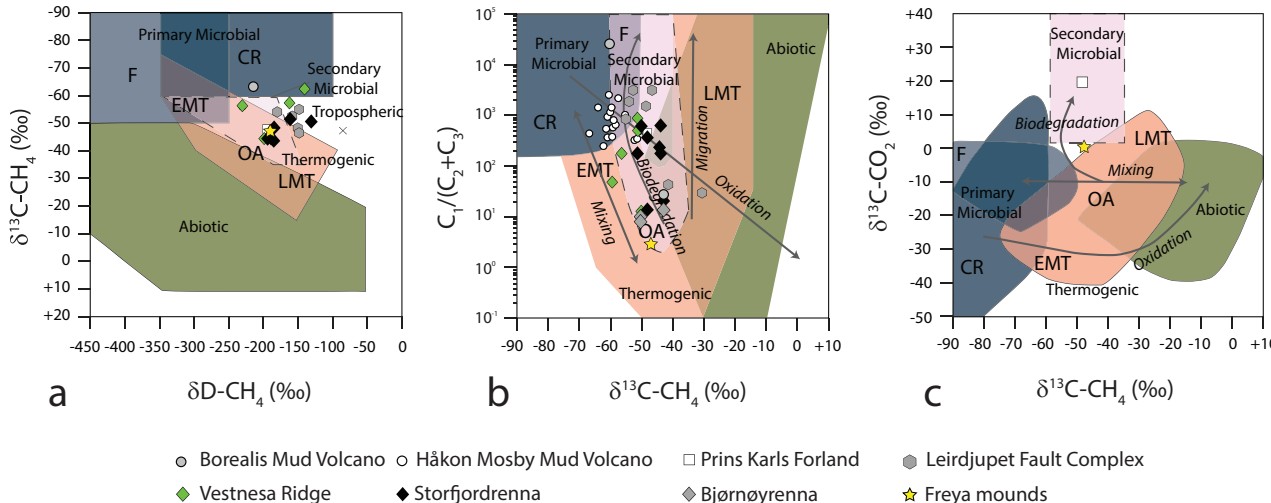

**Fig. 4 | Geochemistry of the gas emitted from Freya gas hydrate mounds.** Molecular and isotopic ($\delta^{13}C$, $\delta D$) composition of the gas contained in the gas hydrate. Sample data from Freya gas hydrate mounds are reported in yellow stars. For comparison, other high-latitudes cold seeps (location in Fig. 1) are reported: Borealis in ref. 15, Håkon Mosby Mud Volcano[38], Prins Karl Forland[37], Leirdjupet Fault Complex[72], Vestnesa Ridge[10], Storfjordrenna and Bjørnøyrenna. Genetic fields of hydrocarbons (CR-$CO_2$ reduction, F−methyl-type fermentation, EMT−early mature thermogenic gas, OA−oil-associated thermogenic gas, LMT−late mature thermogenic gas) after[73]. **a** Isotopic composition of methane. **b** Plot of $\delta^{13}C$-$CH_4$ versus the composition of light hydrocarbon components ($C_1/(C_2 + C_3)$ ratio). Grey arrows indicate the main processes affecting gases' isotopic and molecular compositions. **c** Isotopic composition of $CO_2$ ($\delta^{13}C$-$CH_4$) versus methane $\delta^{13}C$-$CH_4$. The combination of the three plots indicates that the methane in the Freya gas hydrate mounds has a thermogenic origin.

### Table 1 | Taxonomic inventory of fauna collected during ROV dives at the Freya hydrate mounds (79.6 °N, depth 3640 m)

| Phylum | Class | Order | Family | Genus/Species |
|---|---|---|---|---|
| Porifera | Hexactinellida | Lyssacinosida | Rosellidae | *Caulophacus* cf. *arcticus* |
| Cnidaria | Hexacorallia | Actiniaria | Metridioidea (Superfamily) | *Bathyphellia* cf. *margaritacea* |
| | | | | cf *Bathyphellia* sp. |
| | | | | Metridioidea sp. |
| | Staurozoa | Stauromedusae | Lucernariidae | *Lucernaria* cf. *bathyphila* |
| Annelida | Polychaeta | Sabellida | Siboglinidae | *Sclerolinum* cf. *contortum* |
| | | | | *Oligobrachia* sp. |
| | | Scolecida (Infraclass) | Maldanidae | — |
| | | | Capitellidae | — |
| | | Terebellida | Ampharetidae | — |
| | | Errantia (Subclass) | Nephtyidae | — |
| Nemertea | Pilidiophora | Heteronemertea | — | — |
| Mollusca | Bivalvia | Lucinida | Thyasiridae | cf. *Mendicula* sp. |
| | | | | cf. *Rhacothya* sp. |
| | Gastropoda | Litirinimorpha | Rissoidae | — |
| | | Trochida | Skeneidae | — |
| | | Neogastropoda | Buccinidae | — |
| Arthropoda | Malacostraca | Amphipoda | Melitidae | — |
| | | Isopoda | Munnopsidae | — |
| | | Decapoda | | — |
| | Pycnogonida | Pantopoda | Ammotheidae | — |
| Echinodermata | Asteroidea | — | — | — |
| Chordata | Teleostei | Perciformes | Zoarcidae | *Lycodes* cf. *frigidus* |
| | | | | *Lycenchelys* cf. *platyrhina* |

nearby shallow oil seeps of Prins Karls Forland (Fig. 4)[37], based on similarities in age and depositional paleo-environments. For Prins Karls Forland, Arctic blooms of the freshwater *Azolla* fern at 56 Mya[25] and later depositions of organic-rich sediments during the Miocene have been suggested[26,27]. The thermogenic gas contained in the Freya hydrates is distinguished from other known seeps in the

Barents Sea that show a microbial-dominated origin, such as Håkon Mosby Mud Volcano[38], or mixed origin, such as Vestnesa Ridge[10].

Moreover, the observed yellow colour of the hydrates exposed at the seafloor is ascribed to oil-sustaining and/or encrusting bacteria, as observed in the Gulf of Mexico[39]. We incorporated the gas composition of Freya hydrates into a thermodynamic model of the hydrate stability

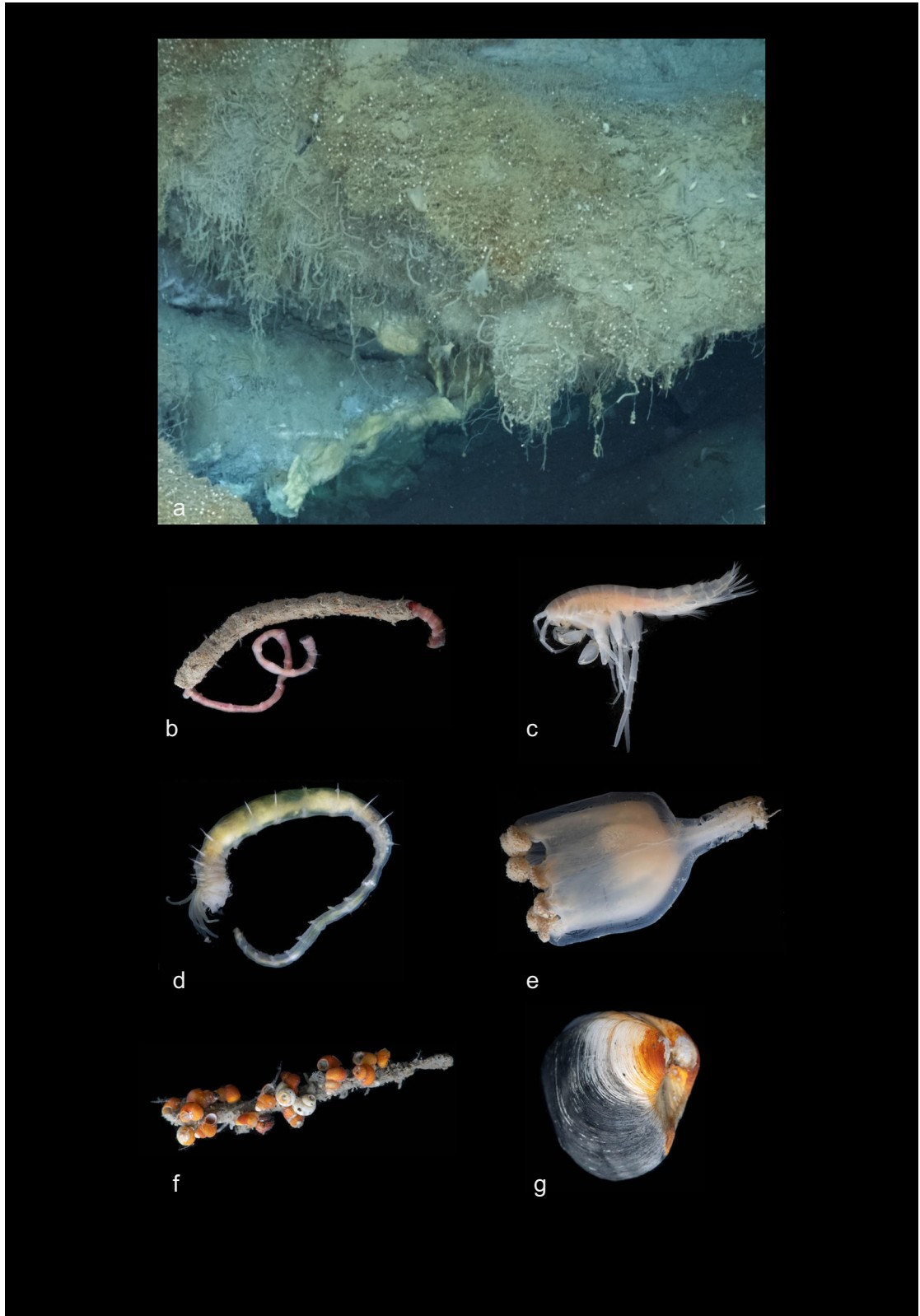

**Fig. 5 | Fauna of the Freya gas hydrate mounds. a** In situ hydrate mound fauna, including *Sclerolinum* forest. **b** Tube-dwelling maldanid polychaete. **c** Melitid amphipod. **d** Ampharetid polychaete. **e** Stauromedusa *Lucernaria* cf. *bathyphila*. **f** Rissoid and skeneid gastropods on a maldanid polychaete tube. **g** Thyasirid bivalve.

zone (see Method section for full parameters). Our model results indicate a subsurface stability zone approximately 248 m thick, suggesting significant potential for gas hydrate accumulation in the sediment. This estimate aligns with previous predictions of a stability zone up to 250 m thick on the flanks of Molloy Ridge[40]. Despite

significant progress in understanding the distribution and concentration of gas hydrates[41,42], a major challenge remains in evaluating gas hydrates as an energy resource and their role in global climate change, resulting from the uncertainty surrounding the size of the resource. In addition, since the 1980s, the Greenland Sea has experienced a

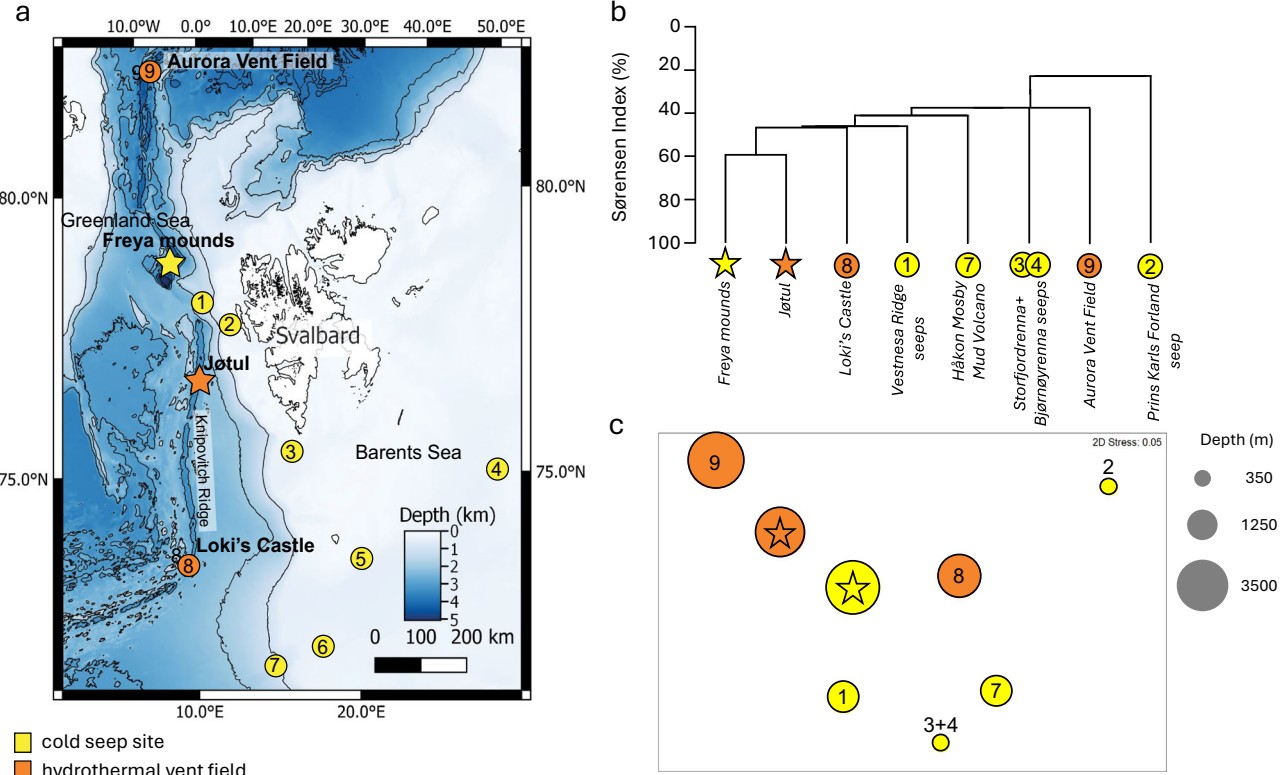

**Fig. 6 | Family-level faunal similarity at high-Arctic (>72 °N) cold seeps and hydrothermal vents. a** Regional map of seeps (yellow) and vents (orange): yellow star = Freya gas hydrate mounds; orange star = Jøtul vent field, sites: 1 Vestnesa Ridge seeps; 2 = Prins Karls Forland seeps; 3 = Storfjordrenna gas hydrate mounds; 4 = Bjørnøyrenna seeps; 5 = Leirdjupet Fault Complex seeps; 6 = Borealis Mud Volcano; 7 = Håkon Mosby Mud Volcano; 8 = Loki's Castle; 9 = Aurora Vent Field. Seabed topography shown is from the Global Multi-Resolution Topography (GMRT) synthesis[70]. **b** Dendrogram of faunal similarity between sites from hierarchical single-linkage agglomerative clustering based on Sørensen Index. Data analysed from this study and published literature for sites (76 families at 8 sites; for data sources, please see Supplementary Table 1; data for Storfjordrenna and Bjørnøyrenna are combined because separate inventories are unavailable in the literature). **c** Two-dimensional ordination of faunal similarity between sites from non-metric multidimensional scaling (nMDS) based on Sørensen Index. Bubble diameters represent site depths (starred yellow bubble = Freya gas hydrate mounds; starred orange bubble = Jøtul vent field).

noticeable warming, with temperatures rising from approximately −1.30 to −0.85 °C by the 2020 s[43]. In the Fram Strait, influenced by both Greenland Sea Deep Water and Eurasian Basin Deep Water, temperatures have fluctuated between −1.20 and −0.95 °C in the 1980s, warming to around −0.85 °C by the 2020s[43]. While we cannot completely rule out the impact of global warming on the Molloy gas hydrate, the complexity of these changes suggests multiple influencing factors and important aspects are associated with the methane's role in supporting local ecosystems

Gas hydrate dissociation contributes to methane seepage into the deep ocean, potentially reaching the upper mesopelagic zone. Where gas hydrates are stable, gas bubbles released are typically coated in a hydrate skin that inhibits their dissolution. Although bubbles lose this coating and dissolve rapidly as they ascend beyond the gas hydrate stability zone[44], the presence of oil can lead to the formation of oil-coated bubbles, which were shown to travel through a 3400 m high water column in the Gulf of Mexico[45]. In ROV video observations (Supplementary Movie 1), we noted numerous trains of bubbles ascending through the water column from localised areas on the seafloor that had visible patches of clear hydrate directly beneath them. Some of these bubbles exhibited unusual flat shapes while rising, which we attribute to the formation of oil and gas hydrate coatings[46].

Water column temperature above the Freya mounds increased from 0 °C at ~1000 m to 2.60 °C at ~300 m depth (Fig. 2), thereby crossing the boundary of the gas hydrate stability zone (~297 m)[47]. The minimum depth at which we observed a bubble plume in multibeam backscatter signals was ~290 m, therefore consistent with gas bubbles losing their hydrate coating and dissolving rapidly as they rise above hydrate stability conditions. The rise height previously reported for a gas flare at this site was ~1770 m (the 'GFA' flare in ref. 32). Our MBES shows a truncated bubble plume at ~3350 m above the seafloor, much higher than this previous measurement and suggesting that the plume reaches even higher levels. It has been previously suggested that methane is generated along the Spitsbergen Transform Fault immediately north of the slow/ultraslow spreading Molloy Ridge and released through boundary faults of the deep sediment-filled Spitsbergen Transform Fault depression[32].

The visual identification of gas hydrate mounds and ridges in different stages of evolution at Freya (from sedimented domes to mounds and arches of exposed hydrate and, finally, pit-like collapse features; Fig. 3) provides a snapshot of distinct features observed simultaneously. While these features are interpreted as representing different stages of evolution, this interpretation is based on their morphology and spatial distribution rather than direct temporal observations. This suggests continual processes of formation and dissociation, consistent with hydrates being dynamic and metastable systems[48].

Hydrate dissociation releases gas and freshwater into the surrounding environment as its crystal lattice breaks down. This may physically disturb fauna that have colonised hydrate mounds, particularly removing substratum occupied by sessile taxa. Availability of methane may also be reduced locally once most or all of the hydrates have dissociated from a structure. The pit-like collapse features that we suggest form where the sedimented mounds have collapsed as a result

of hydrate dissociation have patches of depauperate fauna dominated by taxa such as Stauromedusae and motile species, in contrast to the *Sclerolinum* forest and maldanid tubeworms occupying the mounds of intact hydrates. Hydrate mounds may, therefore, represent a successional deep-sea habitat, with faunal composition changing as a result of disturbance from hydrate dissociation and subsequent waning in methane supply at the individual mound scale.

## Diversity and biogeography of fauna at the Freya gas hydrate mounds

Siboglinid tubeworms are one of the dominant taxa at the Freya mounds and may function as ecosystem engineers, with their tubes providing three-dimensional structure colonised by filamentous bacteria and grazers such as gastropods. Siboglinids have a widespread distribution at other Arctic seep and vent sites (Supplementary Data 1): the frenulate *Oligobrachia* occurs at shallower cold seeps in the region[17], and the monoliferan *Sclerolinum* is the biomass dominant at the Håkon Mosby Mud Volcano[7], also occurs at the Loki's Castle[23], and Jøtul hydrothermal vent fields. *Sclerolinum contortum*, which appears to be the siboglinid at the Freya mounds, may be considered an opportunist chemosynthetic species, exhibiting a cosmopolitan distribution with populations also found at seeps in the Gulf of Mexico and hydrothermal environments in the Antarctic[49].

Maldanid polychaetes, whose tubes also provide an ecosystem engineering function, are also widespread at other Arctic seep and vent sites, along with rissoid snails and thyasirid bivalves (Supplementary Data 1). Several of the taxa present at the Freya mounds demonstrate an overlap in faunal composition between Arctic seeps and vents: melitid amphipods and skeneid gastropods are found at Loki's Castle[23] and the Aurora Vent Field[22], and stauromedusae identified as *Lucernaria bathyphila* are also present at Loki's Castle as well as the background fauna[23].

Chemosynthetic habitat type (vent versus seep) does not segregate sites in family-level faunal composition (Fig. 6). The fauna of the Freya methane seep shows the highest family-level similarity with the fauna at the Jøtul vent field, despite the seafloor environment at Jøtul comprising primarily of basalt and hydrothermal precipitates, contrasting with the sedimented seafloor and sediment-coated hydrates and carbonate structures at Freya. The Jøtul vent field is at a similar depth (3020 m) to Freya (3640 m), however, despite being 266 km away. In contrast, the Vestnesa Ridge seeps are geographically closest to Freya (93 km distance) in our comparison dataset but ~2440 m shallower at a depth of ~1200 m, and show a lower faunal similarity with Freya than the Jøtul vents.

Although there is no correlation between the proximity of sites and faunal similarity, the decline in similarity with increasing differences in depth between sites suggests some depth segregation in the composition of recorded fauna. This has been seen elsewhere in the Arctic and also in other regions for seeps[50]. Some of the animals that we found at the Freya mounds are common in the non-chemosynthetic fauna of the Fram Strait at depths greater than 1000 m, including the stalked sponge *Caulophacus* cf. *arcticus* and the fishes *Lycodes* cf. *frigidus* and *Lycenchelys* cf. *platyrhina*. Several of the taxa reported at shallow Arctic seeps are similarly known from non-chemosynthetic environments at shallower depths, such as abundant populations of snow crabs (*Chionoecetes opilio*) at the Storfjordrenna seeps[51]. The presence of depth-segregated 'background' fauna at Arctic vents and seeps may therefore contribute to the depth-related pattern of faunal similarity in our analysis.

An overlap in chemosynthetic-dependent taxa such as siboglinid polychates at seep and vent habitats in the Arctic may be a particular biogeographic feature of the region[18] as a result of basin geomorphology and recent glaciological history. Firstly, cold seeps and hydrothermal vents occur in close proximity in the Fram Strait, unlike many other biogeographic provinces where continental slopes and seafloor-spreading centres are typically geographically separated. However, where seeps and vents are in close proximity and at a similar range of depths, they tend to share more taxa (e.g. Guaymas Basin[28]). This is especially the case where habitat is similar, such as where vents located in a sedimentary setting occur close to seeps. It is notable that although the Jøtul vent fields were mainly associated with basalts and hydrothermal precipitates, siboglinid tubeworms occurred in microhabitats, such as between rocks or in bacterial mats, where sediment was present[20]. Hydrothermal fluids at the Jøtul vents field also contain a high concentration of methane, perhaps indicating similarities in the biogeochemical environment to the Freya hydrate mounds and other nearby seeps[20].

Secondly, although palaeo-reconstructions differ in estimates for the extent of the ice sheet at the peak of the Last Glacial Maximum ~20,000 years ago, large areas of ocean were covered by a floating glacial ice shelf up to 1 km thick[52,53]. Such thick glacial ice cover would have blocked out sunlight for photosynthesis in the underlying ocean, reducing phytodetrital flux to the deep seafloor in a manner similar to parts of the Arctic that were permanently covered in multiyear sea ice until recently, even despite the much thinner and light-permeable cover provided by sea ice compared with floating ice shelves of meteoric ice. This is known to lead to depressed abundance and diversity of the benthic fauna because of low food supplies in the deep sea. The greater spacing of vent fields along the ultraslow-spreading ridges of the Arctic[54] may also have favoured chemosynthetic-dependent taxa that can also colonise cold seeps, leading to an overlap in fauna between the two habitat types in the region through stepping-stone dispersal.

The chemosynthetic-dependent taxa at Arctic vents and seeps appear to conform with Thorson's Rule, which predicts an absence of species with planktotrophic larval development at high latitudes[55]. Taxonomic groups with planktotrophic development that are widespread at low-latitude vents and seeps, such as bathymodiolin mussels and alvinocaridid shrimps, have not been recorded at Arctic or Antarctic chemosynthetic habitats[56]. The reduction in phytodetrital flux into the Arctic deep sea at the Last Glacial Maximum may have favoured the taxa with non-planktotrophic development found at Arctic vents and seeps, such as melitid amphipods and siboglinid polychaetes[57].

The Freya mounds represent the first cold seeps found at a depth comparable with hydrothermal vents in the Arctic, and future taxonomic studies beyond family-level identifications will elucidate whether species at Freya are the same as those found at Arctic vents or represent further undescribed species. Ongoing identifications indicate shared genera between the Freya hydrate mounds and Jøtul vent field, including the melitid amphipod *Exitomelita* and *Skenea* and *Rissoa* gastropods. At present, there is an inevitable disparity in sampling effort between recently discovered sites such as Freya and Jøtul and longer-studied sites such as Loki's Castle and Håkon Mosby Mud Volcano, reflected in the extent of their faunal inventories (Supplementary Data 1). Further sampling and more detailed work, including genetic barcoding of specimens for each taxon from each site, will be required to resolve biogeographic relationships of Arctic chemosynthetic habitats at the species level.

In April 2024, the Norwegian government opened an area in Norway's extended Exclusive Economic Zone between Jan Mayen and Svalbard for deep-sea mining activities[58], and although initial licensing of areas for mineral exploration was paused in December 2024, its future development is anticipated. The discovery of the Freya hydrate mounds and their associated fauna highlights the need to understand the composition and distribution of species and deep-sea habitats across this region to develop robust, evidence-based regional environmental management plans that minimise risk of biodiversity loss and impacts on Vulnerable Marine Ecosystems such as active seeps and vents or adjacent sponge fields and stalked crinoid communities[59]. Mining activities impacting active vent habitats are inconsistent with

international obligations to protect biodiversity[60], and the overlap between vent and seep fauna in the Arctic indicates that cold seeps may need protection similar to that recommended for active hydrothermal vents to preserve the diversity of chemosynthetic fauna in the region.

## Methods

All the data and samples analysed in this paper were acquired from the RV *Kronprins Haakon*, using vessel equipment and the ROV *Aurora*. Data and samples were collected at the Jøtul vent field on 12–13 May 2024 and at the Freya gas hydrate mounds on 18 May 2024. The biological specimens collected during the expedition are deposited in Tromsø, at the Department of Geosciences, UiT The Arctic University of Norway, where they are curated in accordance with the institutional and Ocean Census guidelines. Sampling and research activities were conducted under a permit (number 24/4594) issued by NOD, the Norwegian Offshore Directorate, on 26/04/2024.

### Sonar data acquisition

Seafloor mapping and water column investigations were performed using a hull-mounted Kongsberg EM302 1 × 1° MBES system on the *RV Kronprins Haakon*. The MBES operates at frequencies of 26–34 kHz and has a depth range of 10–8000 m. The Kongsberg EM302 data were processed onboard for bathymetry, backscatter and water column anomalies. We also detected gas seeps using the ROV's Norbit sector-scanning search sonar, confirming the source of the ship-detected gas flares.

The bathymetry was processed using the QPS Qimera software, and the gas flares were detected using the QPS FMMidwater software from the backscatter signal. This software converts the *.all and *.wcd files obtained from the EM302 into generic water column format (*.gwc) files, which can be modified to target specific beams where the flares are visible. The selected flares are then exported as sd files and imported into Fledermaus for visualisation along with the processed bathymetry.

### Hydrographic profiling

Water column profiles of temperature and salinity were measured using a Seabird 911 Plus CTD (Conductivity-Temperature-Depth) probe fitted with dual SBE3 temperature sensors and dual SBE4 conductivity sensors. The CTD 104 (79.6137 °N 3.6552 °E) was lowered from the ship at a winch speed of 60 m per minute to an altitude of 12 m above the seafloor, detected by a Teledyne PSA-916 acoustic altimeter. Data were recorded via an SBE 11plus V2 Deck Unit and processed using SBE Data Processing software version 7.26.7 into 1 m depth-average values.

### ROV Aurora video survey and sampling

The REV Ocean ROV *Aurora* is a Kystdesign Supporter ROV capable of diving to 6000 m. *Aurora* was deployed through the moonpool of RV *Kronprins Haakon*, which enabled operations in sea ice. The two cameras used for science were a SubVIS Orca, an IP Zoom HD Camera with an optical zoom of 30×, and a SubC Rayfin Mk2 Benthic 4k camera with a digital zoom equivalent to a 5× optical zoom. Two parallel lasers 0.16 m apart provided a scale in images. Samples were obtained during ROV Dive 17 and 19 (18 and 19 May 2024, respectively) using 0.3 m pushcores, a blade corer (a rectangular blade that cuts into the sediment, allowing for a clean entry and exit and preserving the stratification and structure of the sediment), a suction sampler with 8 chambers on a rotating carousel, a scoop, and direct collection by the ROV's manipulator arm. All the samples used for this biological study were collected in the vicinity of a hydrate mound located at 79.6143 °N 3.6563 °E.

### Processing of faunal samples and data

Upon arrival on deck, biological samples were sieved and cleaned in filtered seawater. Fauna specimens were live-photographed with macrophotography equipment consisting of a Nikon D6 and Nikon D850 with AF-S Micro Nikkor 105 mm 1:2.8G ED and Sigma 50 mm f/1,2 DG DN Art L-mount lenses, and ProFoto B2 Portable Flash. The specimens were identified to the lowest taxonomic level possible aboard, then preserved in 96% ethanol, or 4% formaldehyde buffered with borax, or frozen at −80 °C for future taxonomic and ecological analyses.

Taxa identified to family level from Freya and Jøtul by our expedition were compared with family-level presence/absence data compiled from published literature for other seeps and vents in the region (76 families from 8 sites in total; details of sites and data sources are presented in Supplementary Data 1)[61–63]. Taxa that were not identified to family level in our samples or in published inventories of fauna were excluded to avoid possible exaggeration of faunal similarity from conflation of higher-rank taxa shared between sites. An all-pairwise similarity matrix between the seep and vent sites was calculated from the family-level presence/absence data using the Sørensen Index[64]:

$$S = \left[ 2n_{ab} / (n_a + n_b) \right] \times 100 \qquad (1)$$

where $S$ is the Sørensen Index value (%) of faunal similarity; $n_{ab}$ is the number of taxa shared between two sites $a$ and $b$; $n_a$ is the total number of taxa at site $a$; and $n_b$ is the total number of taxa at site $b$. Non-metric multidimensional scaling and hierarchical single-linkage agglomerative clustering were applied to the similarity matrix using PRIMER v7 software to generate a two-dimensional ordination and dendrogram of faunal similarity relationships[65].

To compare variation in faunal similarity with variations in depth and the geographic separation of sites, matrices were constructed for pairwise depth differences and distances between sites. The distances between sites were calculated from latitude and longitude values using the haversine formula for great-circle distances ($d$):

$$a = \sin^2(\Delta\varphi/2) + \cos\varphi1 \cdot \cos\varphi2 \cdot \sin^2(\Delta\lambda/2) \qquad (2)$$

$$c = 2 \cdot \text{atan2}\left(\sqrt{a}, \left(\sqrt{1-a}\right)\right) \qquad (3)$$

$$d = R \cdot c \qquad (4)$$

where $\varphi$ is latitude, $\lambda$ is longitude, and $R$ is the mean radius of the Earth (6371 km). Spearman rank correlations were then used to compare the faunal similarity matrix with the matrices of depth differences and distances between sites.

### Hydrate and sediment-bound gas geochemistry analysis

A gas hydrate sample was collected by the ROV manipulator using a blade corer (at 79.6143 °N 3.6624 °E) (Supplementary Fig. 1). Once the blade corer was on deck, to minimise the gas hydrate dissociation that started already when the ROV was ascending because of change in pressure and temperature, we immediately opened the lid and collected four replicates of gas hydrate using a sterile syringe. We transferred them to 20 mL glass vials, which were sealed with a rubber septum and crimp cap. Similar methods have proved to be effective for sampling gas hydrate[66].

Two sediment samples of known volume (5 mL) were extracted from a blade core (4 cm and 9 cm), transferred to glass vials containing 5 mL of 1 M NaOH and stored upside-down at 4 °C until headspace gas analyses. Headspace gas analyses were conducted at Applied Petroleum Technology (APT) laboratories in Oslo, Norway. Aliquots of gas for molecular analyses were injected into an Agilent 7890 RGA GC equipped with Molsieve and Poraplot Q columns and measured on a flame ionisation detector (FID). Hydrocarbons were measured by FID. The carbon and hydrogen isotopic composition of methane were

determined on a Trace 1310 gas chromatograph (Thermo Fisher Scientific), equipped with a Poraplot Q column and PTV (Programmable Temperature Vaporizing) injector. The GC was interfaced via GC-Isolink II and Conflo IV to a Delta V Isotope Ratio Mass Spectrometer (Thermo Fisher Scientific). Precisions on $\delta^{13}C$ and $\delta D$ were better than 1‰ vPDB (2 s) and 10‰ vSMOW (2 s), respectively.

## Oil geochemistry analysis

Oil analyses were conducted on hydrate-derived samples and from a sediment sample. All oil preparation and analysis procedures followed NIGOGA (Norwegian Industry Guide to Organic Geochemical Analysis), 4th Edition, and were conducted at APT (Oslo). Samples were extracted in approximately 80 cc of dichloromethane with 7% (vol/vol) methanol. An aliquot of 10% of the extract was transferred to a pre-weighed bottle and evaporated to dryness. For deasphaltering, extracts were evaporated almost to dryness before a small amount of dichloromethane (three times the amount of Extractable Organic Matter, EOM) was added. Pentane was added in excess (40 times the volume of EOM/oil and dichloromethane). The solution was stored for at least 12 h in a dark place before the solution was filtered/centrifuged. Gas chromatographic analyses of the EOM and saturated fractions were performed using an HP Agilent 7890A GC Gas Chromatograph equipped with a CP-Sil-5 CB-MS column, length 30 m, i.d. 0.25 mm, film thickness 0.25 μm. Saturated and aromatic fractions were analysed via GC-MS using a Thermo Scientific DFS™ magnetic sector mass spectrometer. The instrument was tuned to a resolution of 3000, and data were acquired in Selected Ion Recording (SIR) mode. The column used was a 60 m CP-Sil-5 CB-MS with an i.d. of 0.25 mm and a film thickness of 0.25 μm.

## Modelling of gas hydrate stability

The gas hydrate stability zone was calculated for a depth of 3640 m by assuming steady state conditions and applying in situ values of bottom-water temperature of −0.6 °C, salinity of 35 PSU, and the geothermal gradient of 120 °C/km reported by ref. 67 for Molloy Deep of 120 °C/km. The model was implemented on CAGEHYD software[68] based on the CSMHYD code[69]. The model was run for hydrates having the molecular composition measured in our study.

## Reporting summary

Further information on research design is available in the Nature Portfolio Reporting Summary linked to this article.

## Data availability

All the data generated and or analysed in the study are included in the main text and in the Supplementary Information file. The species identified at the Freya gas hydrate mounds, the ROV frame showing the Freya gas hydrate mounds and the gas hydrate sampling on which the biological and geochemical analyses for this paper, the n-Alkanes chromatograms of the oil from Freya gas hydrate mounds, the source rock proxies, the oil maturity proxies, all-pairwise faunal similarity matrix (Sørensen Index values) for high-Arctic (>72 °N) cold seeps and hydrothermal vents, calculated from family presence/absence data using faunal records from this study and published literature (see Supplementary Data 1 for data sources), and the Fledermaus plot showing where the flares originate on the topography as processed with Qimera and acoustic backscatter processed with FMMidwater using the shipboard MBES at the Freya gas hydrate mounds are provided in the Supplementary Information. A video showing methane bubbles in this study is provided as Supplementary Movie 1.

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

## Acknowledgements
UiT The Arctic University of Norway, Ocean Census, and REV Ocean supported our research and the Deep Arctic – EXTREME24 expedition. Ocean Census acknowledges the funding of The Nippon Foundation, which supported this expedition. K.L. is part of the British Antarctic Survey Polar Science for Planet Earth Programme and funded by the UK's Natural Environment Research Council. The RV Kronprins Haakon officers and crew, the ROV Aurora team, and Steve Killops are deeply acknowledged.

## Author contributions
G.P.: conceptualisation, methodology, writing, original draft preparation, funding acquisition. J.T.C.: conceptualisation, methodology, writing. K.L.: methodology, writing. V.N.: methodology, writing. E.R.-L.: methodology, writing. C.A.: data visualisation, methodology. B.F.: data visualisation, writing. A.D.R.: conceptualisation, methodology, writing, funding acquisition. AD-E24 team: methodology.

## Funding

## Competing interests
The authors declare no competing interests.

## Additional information

## the Arctic Deep - Extreme24 consortium

**Alejandra Saenz de Tejada[6], Alex David Rogers[4], Alfredo Rosales Ruiz[7], Asgeir Steinsland[8], Carlotta Redaelli[1,9], Clarisse Goar[10], Daniel Despujois[6], Eva Ramirez-Llodra[5], Ewan McEvoy[11], Fereshteh Hemmateenejad[9], Giuliana Panieri[1], Ida Søhol[12], Ines Barranchea Angeles[1], Jack Hogan[13], Jessica Michelle Webster[14], Joe Sharman[15], Jonathan T. Copley [iD][2] ✉, Katrin Linse[3], Laura Warmuth[16], Lawrence Hislop[5], Leif Johan Ohnstad[8], Leighton Rolley[5], Martin Hartley[15], Nuria Rico Seijo[13], Pamela Rivadeneira[17], Patricia Esquete Garrote[18], Patrick Vågenes[5], Pedro Furtado Costa Rodrigues[14], Raissa Hogan[19], Stig Vågenes[5], Tor-Arve Lunde[5], Usha Parameswaran[20], Verity Nye[15] & Will West[15]**

[6]CIIMAR Terminal de Cruzeiros do Porto de Leixões, Matosinhos, Porto, Portugal. [7]Fundación Museo del Mar de Ceuta, Ceuta, Spain. [8]Institute of Marine Research, Bergen, Norway. [9]Department of Earth and Environmental Sciences (DISAT), University of Milano Bicocca, Milan, Italy. [10]IFREMER, Plouzané, France. [11]School of Biological and Environmental Sciences, Liverpool John Moores University, Liverpool, UK. [12]Department of Arctic and Marine Biology, The Arctic University of Norway, Tromsø, Norway. [13]Nekton, Begbroke Science Park, Oxfordshire, UK. [14]BBC Natural History Film Unit, Bristol, UK. [15]The Nippon Foundation-Nekton Ocean Census Programme, Begbroke Science Park, Oxfordshire, UK. [16]Department of Biology, University of Oxford, Oxford, UK. [17]Laboratorio de Ecosistemas Costeros, Plataforma y Mar profundo, Museo Argentino de Ciencias Naturales "Bernardino Rivadavia" (CONICET), Buenos Aires, Argentina. [18]Departamento de Biologia & CESAM (Centro de estudos do Ambiente e do Mar), Universidade de Aveiro, Aveiro, Portugal. [19]School of Natural Sciences, University of Galway, Galway, Ireland. [20]Centre for Polar Ocean Research (NCPOR), Ministry of Earth Sciences, Government of India, Vasco-da-Gama, Goa, India.

