## [Peer Review file · Nature Communications]

Deep-sea gas hydrate mounds and chemosynthetic fauna discovered at 3640 m on the Molløy Ridge, Greenland Sea

Corresponding Author: Professor Giuliana Panieri

Version 0:

Reviewer comments:

Reviewer #1

(Remarks to the Author)

Although the discovery of the deepest yet discovered chemosynthetic community is rather surprising and of interest to the scientific community, the manuscript has significant deficiencies that render it unsuitable for publication in Nature Communications. For example, the caption for Figure 4 does not match the figure, instead it is for Figure 5. Of most concern is the sediment coring and sampling methods. Gas exsolution and gas hydrate decomposition were not mentioned and it is surprising the authors did not observe rapid degassing and sediment extrusion once the cores arrived on deck. The anomalously high 85.9 mM concentration of methane in a surface seafloor sample does not make any sense, unless the sample was collected using a pressure core barrel. Thus, the gas analyses are not representative of the in situ concentrations as plotted in Figure 5 and it is not clear if the units of methane concentration are for the sediment pore fluid. The discussion of gas hydrate structures in lines 275-286 is not relevant to the manuscript and is speculative. There is a substantial amount of speculation and inference throughout the text which is not supported by the data collected. For example, lines 281-282; 305-312; and 326-331. The isotope and C1/C2+C3 data can be summarized in a sentence or two to establish the general source of the gas, but going beyond that and also including a 4 panel Figure 4 is not relevant to the manuscript.

Reviewer #2

(Remarks to the Author)

Review of:

Unveiling seafloor gas hydrate mounds with chemosynthetic fauna at 3640 m deep in the Molløy Ridge, Greenland Sea by Warren Wood

<https://mts-ncomms.nature.com/cgi-bin/main.plex?el=A2S6DUm7A6HOjR5J2A9ftdbqWYHICFPii28YUwrZxzQQZ>

Recommendation: publish with minor revision

This manuscript describes data acquired around a methane hydrate mound in unprecedented water depths, and analysis showing that fauna whose taxa favor those found at a hydrothermal vent. This is truly a remarkable find - the deepest hydrate mound (3640 m deep) with the tallest known methane plume in the water column. Also remarkable is the observation that fauna with similar taxa are being nourished by significantly different sources. These findings should be published, and Nature Communications is the appropriate publication.

This manuscript contains significant qualitative description, which in my opinion is warranted given the unusual depth of this hydrate mound complex and the similarity in associated fauna to hydrothermal vents.

However, I would prefer a sharper conclusion. Figure 7 is the ideal place to make this point, but the data displayed here could be far better focused. At least a box around the Freya and Jotul site data (with labeling to indicate cold seep vs. hydrothermal vent) highlighting the faunal similarity between two vastly different types of seepage. It's OK to be a little redundant when conveying the point of the manuscript.

I would also prefer a discussion that is more focused on supporting support the main conclusion. The authors have an important point to make, but it is not too complicated, and this is not a "long-format" article. The discussion surrounding the main point (descriptions of other sites, and details of the fauna) are necessary, but their purpose is to provide context that should support the main point. To that end it is not clear that the manuscript requires both figures 4 and 5 perhaps one would suffice.

Minor comments:

Figure 3 caption – It might be over interpreting the image to call it a gas blow out. Unless there is supporting evidence, it could just as well be a manifestation of a slow dissolution event.

Lines 95-97, regarding depths of seeps and vents

This seems inconsistent with the previous text describing hydrothermal venting in the 500-750m depth range, and cold seeps at 70-800 m water depth. Or are the cold seeps at similar depths significantly far away, or oceanographically removed?

Lines 351-356

I do not see the explanation for how the shapes and morphologies of the hydrate mounds indicates their stage of evolution, or more specifically, the stage of population evolution the authors state. It seems that the nutrients that sustain the fauna might be affected by sediment cover, but not necessarily the shape of the mound.

Figure 7 – might be helpful to repeat the color and symbol legend, to better distinguish between the cold seeps and hydrothermal vents.

Lines 399-400 Could modern (last few hundred years) ocean currents be responsible for similarities along depths instead of across depths?

Note to the editor as well as authors: Lines 424-434 describe the political/environmental/legal impacts of the scientific results, and a context within which they are viewed. I have no expertise to comment intelligently on "international obligations", or if it should even be discussed in this manuscript.

Reviewer #3

(Remarks to the Author)

Panieri et al. present results describing a newly discovered gas hydrate field along the Molløy Ridge at 3640m. I commend the authors on their discovery; however, the paper needs a major revision before it can be considered for publication in Nature Communications. In the following pages, I present comments/questions by line number. I hope that my comments help the authors improve the manuscript.

Detailed comments for the Authors

Line 28: Here and elsewhere in the manuscript: Gas hydrates do not fuel methane seepage. Quite the contrary, methane seepage, when occurring within an appropriate T-P window, fuel methane hydrate formation. This sentence should be re-written along the lines of "Seafloor seepage of methane can lead to the formation of methane hydrate. Methane seepage also supports chemosynthetic communities in the deep sea."

Line 30: Suggest changing to: "...vents found at greater depths along and near active spreading centers". Note: Hydrothermal fluid discharge is not limited solely to mid-ocean ridges. Substantial off-axis fluid discharge occurs in both the Atlantic (e.g., Lost City) and in the Pacific (e.g., YBW-Sentry).

Line 43: It is worth including oil production activities in the last sentence "...deep sea mining and oil production activities in the region"

Line 52: In the Arctic, structure II gas hydrates are actually stable at much shallower depths at ultracold temperatures (-2-C; see the work in the Kara and Laptev Seas).

Line 59: The Milkov 2004 reference for the size of the gas hydrate reservoir is out of date. A more recent estimate is available in: Lee, M.W., B.J Phrampus, A. Skarke, and W.T. Wood, 2022. Global estimates of biogenic methane production in marine sediments using machine learning and deterministic modeling. *Global Biogeochemical Cycles*, <https://doi.org/10.1029/2021GB007248>. Though this paper focuses on biogenic methane production, they also estimate global stores of gas hydrate -- "Our estimated range, as indicated by error bars, of carbon sequestered in hydrate is 62–6,951 Gt." Given the robust modeling approached applied by Lee et al., this paper is worth mentioning.

Line 62: Biogenic methane has also been documented in the Gulf of Mexico, see Klapp et al. 2010, *Marine Petroleum Geology*, <https://doi.org/10.1016/j.marpetgeo.2009.03.004>.

Line 64: This sentence is misleading and confusing to the non-expert. Here again, gas hydrate is defined as a type of cold seep. That is wrong: A methane cold seep can exist without gas hydrate, BUT gas hydrate cannot exist without a source of

methane via methane seepage. It is the oxidation of methane and the coupling of AOM to sulfate reduction, that produces the sulfide that fuels many of the chemo-symbioses found at methane seeps. However, the sulfide is not only from coupling to AOM, oxidation of dissolved organic matter and higher alkanes also fuels sulfate reduction and sulfide production. Please make this clear.

Line 106: This sentence is poorly constructed. You are trying to make too many points and the reader will be confused: Arctic blooms...later depositions of organic-rich sediments during the Miocene potentially created seabed hydrocarbon reservoirs...". First, organic-sediments do not create hydrocarbon reservoirs. Second, is it relevant that the sediments contain freshwater (Azolla ferns) and other materials? Please clarify and focus this sentence.

Line 114: Seismic data cannot provide any information on the source of gas (biogenic vs. thermogenic); please clarify.

Line 117: Simplify - The source of these pelagic plumes have not yet been identified.

Line 125: Hydrate does not support chemosynthetic fauna, dissolved gas (and likely DOM) seepage fueling sulfate reduction does.

Line 145: You are reporting contiguous gas flares that reach from ~3600m to ~300m. The amount of gas/methane discharge required fuel flares 3300m tall would be tremendous. Does the ROV imaging of gas discharge at the seafloor support this. Rather than showing 2D images, I would like to see Qimera or Fledermaus 3D renditions of the plumes. Perhaps these hydrates sit atop a high-flow fault but more typically, deep (high pressure) seeps trap methane efficiently in hydrate. There is always some leakage around the hydrate, but for there to be sufficient discharge to fuel the enormous plumes you report, something unusual has to be going on. Do you have an explanation?

Line 155: It would be extremely unusual for "gas seepage rising from the mounds". First, this infers that the mounds are cohesive and the images show that they are not intact mounds. Rather they are horseshoe shaped or they have a cavern (e.g., Fig. 3). Intact hydrate mounds block and effectively reduce methane seepage, such that any methane escape occurs along the edges of the hydrate structures. Please clarify.

Line 173: I am curious, why is the hydrate greenish-yellow? I have seen structure 1 hydrate with a blue-green tinge and I have seen yellow grading to orange hydrate (oil inclusion) but I do not recall ever seeing or reading a report of greenish-yellow hydrate?

Line 178: The alkane distribution in the sampled hydrate is quite typical for structure II hydrate. It is not "exceptionally" wet. The alkane composition is similar for other sites (for example the Gulf of Mexico), where structure II hydrate abounds.

Line 189: In the text, you report a methane concentration of 85.9 mM in surface sediments. It is simply not possible to measure such a high concentration unless special (a pressure corer) instruments are used. Else, the gases come out of solution during recovery. Furthermore, this high concentration - 85.9 mM - is not shown on Figure 5. Please clean up / correct.

Line 223: The black precipitate is likely iron sulfide. Did you check to see if it was acid volatile?

Line 271: There is certainly a potential for gas hydrate destabilization in the future, especially as the ocean warms. But, do you really expect the Arctic to warm at >3000m? I find that hard to believe. To make such a statement, you need to include evidence from physical oceanographic models showing deepwater - extremely deepwaters - are likely to warm. Else this is simply speculation and given the importance of this topic, the scientific community needs to be standing on firm ground and the way this is presented it is not.

Line 287: What is "Type 4"? This is an older paper and we have learned much since it was published. Hydrates often form along faults beneath the sediment. As gas seepage continues, hydrates are literally pushed upwards. Hydrates are buoyant and the sediment drape can function as an anchor. As hydrates move above the sediment-water interface, they begin to sublime, releasing dissolved methane (not so much bubbles - bubbles are emitted along the edges). As they age, they dissolve and caverns can develop, as can carbonate casts. Again it's the methane seepage - not the hydrate - that drives the community. Without gas seepage, the hydrate, and the chemosynthetic community, would not exist.

This part of the discussion is extremely speculative and needs to be anchored in more data showing specific examples and references. Without that, the "spongy" description needs to go. ROV video should be included to confirm the statements about pelagic bubbles and their source. The authors have not shown enough data to tell a story about "hydrate evolution". What you have is a snapshot that shows frequent hydrate, and this hydrate, I presume, occurs at the surface expression of a fault. A geological map and/or subbottom data would strengthen this argument. Hydrate evolution in other systems, like the Gulf of Mexico, is well documented. Please cite that literature.

Version 1:

Reviewer comments:

Reviewer #1

(Remarks to the Author)

The authors are reporting on a remarkable discovery of methane-fueled seep communities at an unexpected water depth of ~3640m. They have more than adequately addressed the reviewer's comments except for their discussion of the type of gas hydrate found at the seep site. They infer that the gas hydrate is a Structure II hydrate based on the gas chemistry (lines 292-302). This is not relevant to the paper and I suggest removing this section. An inference would easily be misunderstood by readers that Structure II gas hydrate was found at this site, which is not the case. Laboratory confirmation of the gas hydrate structure is necessary before making a statement about structural type. In lines 302-304 an estimate of the depth of gas hydrate stability in the sediment is provided. Please include what geothermal gradient was assumed in this model. The comprehensive faunal comparison provided in Table 2 would be greatly improved by implementing the following suggestions: (1) arrange the columns in order of decreasing depth from left to right; (2) use larger and different symbols for vent vs. seep designations for each site - either shape or color; (3) highlight the chemosynthetic-dependent taxa in the family column. Except for specialists, most readers will not easily recognize the families of these taxa. I have attached my version of a reorganized Table 2 to this review. I am not sure if the depth and spatial arguments are strongly supported by the data presented in this table. The issues I see, which are inherent in a discovery like this and a comparison to other regionally relevant chemosynthetic communities is that they are under-sampled. Paucity of representative organisms may be due to numerous factors, such as limited and selective sampling, thus the collection of organisms is not representative. I think it is difficult to make convincing generalizations at this point. Please re-examine the discussion in light of what Table 2 shows, discuss its limitations, and modify the text as appropriate. There are site labeling inconsistencies between Figures 1 and 6. The numbered sites do not correspond with each other. The Figure 1 legend does not describe sites 8 and 9 on the map. Sites names are not consistently located. Please label the maps in the same way and correctly name the sites in the figure captions. In the caption for Figure 3, the word 'depressions' is used. I think 'collapse features' is a more accurate description. Please change 'depression' to 'collapse feature' throughout the paper. In addition, there is no evidence for a 'gas blowout'. This implies evidence has been observed for a significantly pressured gas pulse that created the morphologic feature. Please use a more descriptive term that does not imply process. Line 643 - 'Borealis in [13]' does not make sense. Please correct. In Figure 2, the lines are not dashed as mentioned in the caption. Please correct the figure. The statements in Lines 99-103 are awkward and it is difficult to understand the point(s) that are being made. Line 284 - I think 'first order' would be clearer than 'first step'.

Reviewer #3

(Remarks to the Author)

In this revision Panieri et al. have refined their results describing a newly discovered gas hydrate field along the Molløy Ridge at 3640m. Despite their extensive revision, some issues remain that must be resolved before the paper is considered for publication in Nature Communications.

Detailed comments for the Authors

For the distribution of structure II gas hydrates in the Arctic, please see -- Reviews of Geophysics
The interaction of climate change and methane hydrates, Carolyn Ruppel & John Kessler, 2016.

I made this comment on the original paper and my concern remains. "You are reporting contiguous gas flares that reach from ~3600m to ~300m. The amount of gas/methane discharge required fuel flares 3300m tall would be tremendous. Does the ROV imaging of gas discharge at the seafloor support this. Rather than showing 2D images, I would like to see Qimera or Fledermaus 3D renditions of the plumes. Perhaps these hydrates sit atop a high-flow fault but more typically, deep (high pressure) seeps trap methane efficiently in hydrate. There is always some leakage around the hydrate, but for there to be sufficient discharge to fuel the enormous plumes you report, something unusual has to be going on. Do you have an explanation?"

The reviewers added a video, but the discharge shown in the video is not consistent with the description of the acoustic flare shown in Fig. 2. It is strange that no dimensions are given for the flare in Fig. 2. Based on my experience mapping methane seeps in the Atlantic and Pacific, the bubble discharge shown in the video is inconsistent with the type of discharge required to generate a plume that reaches thousands of meters above the seafloor. A more sophisticated analysis of the geophysical data is needed to generate a cohesive, convincing argument.

I made the following comment on the original manuscript "I am curious, why is the hydrate greenish-yellow? I have seen structure 1 hydrate with a blue-green tinge and I have seen yellow grading to orange hydrate (oil inclusion) but I do not recall ever seeing or reading a report of greenish-yellow hydrate?"

The authors provided this resource as evidence of "greenish hydrate": <https://geoexpro.com/gas-hydrates-part-i-burning-ice/> in a text written by Lasse Amundsen and Martin Landrø, published Date: June 12, 2012. This reference shows a green-colored methane molecule inside a cage of yellow water molecules. The cartoon has nothing to do with the color of hydrate observed by the authors. The "green" color could be glauconite and the yellow could be organic sulfur. It is also possible/likely that yellow-colored Beggiatoa could be associated with hydrate. This point needs more clarification.

In the original manuscript, I commented "There is certainly a potential for gas hydrate destabilization in the future, especially as the ocean warms. But, do you really expect the Arctic to warm at >3000m? I find that hard to believe. To make such a statement, you need to include evidence from physical oceanographic models showing deepwater - extremely deepwaters - are likely to warm. Else this is simply speculation and given the importance of this topic, the scientific community needs to be standing on firm ground and the way this is presented it is not."

The author's response is unsatisfactory. Even if the bottom water warmed to +1°C (or more), hydrate would be stable at the extreme depths of this site given the pressure. Arguing that the hydrate will be unstable if deepwater warms to -0.85°C is baseless and shows a lack of understanding of hydrate dynamics in the natural environment.

Regarding the last point about the hydrate being "spongy" - hydrate is never spongy. Solid hydrate is quite firm. Looking at Supp Fig 1, the "hydrate" appears like a combination of a sponge (actual biological organism!) and a Beggiatoa mat. The blade core sample is not convincing - the filaments are strange and not like any hydrate I have ever seen. I am left more confused than convinced by the additions to the paper.

Reviewer #4

(Remarks to the Author)

The manuscript describes the discovery of the world's deepest known seafloor hydrate mounds and associated cold-seep fauna with taxa similar to hydrothermal vent in the area rather than to other, shallower cold seeps. The author also report contiguous gas flares extending from approximately 3600m to 300m and the proposed mechanisms deserve publication on their own, as they demonstrate the huge contribution that ultra-deep methane systems can make to global methane cycling under specific conditions. These findings are of great interest for a vast community and should be in Nature Communications.

In reviewing the manuscript I also kept into consideration the previous round of revision, including the comment made by three reviewers and the work done by the authors to respond to the comments. Overall I believe that the authors addressed all the reviewers previous comments and that the manuscript is currently in great shape. I appreciate that the authors briefly discuss the wider context of results in term of environmental management plans. More should be done in this direction when findings can potentially impact ecosystem and resource management.

A few minor comments follow:

line 66. maybe worth mentioning that the chemosynthetic production is of prokaryotic origin since the rest of the paragraphs focuses on eukaryotes? Being Nat Comm a generalist journal the author should not assume that all the readers are familiar with chemosynthesis and the current paragraph might be misleading.

Line 99-103. Some of the differences in the fauna between vents and cold seeps might also be driven by the different geochemical sources (and temperature regimes) supporting very diverse microbial communities in these ecosystems. Perhaps this could be mentioned citing relevant microbiological literature.

Line 249-254. As s suggestion, an additional way to test for this is using a matrix of beta diversity between sites and testing its dependence on the pairwise distance between sites using a mantel test.

Line 351-361. I agree with a previous reviewer that the current study did not classify the morphological evolution of a site, rather describes a series of snapshot from diverse features in the site with the assumption (according to my best understanding) that they represent different time-points of morphological evolution. However, they are still distinct features observed at the same time point, and this can be further clarified also in line 134-1367.

Line 406-420- Very interesting section of the discussion.

Version 2:

Reviewer comments:

Reviewer #1

(Remarks to the Author)

This manuscript has matured during the review process and I recommend publication after a few important corrections and clarifications are made.

1. Abstract - lines 35-36: Please clarify by what is meant by 'display morphologies resulting from progressive stages of gas hydrate dissociation'. Are the morphologic changes, changes in the assemblage of taxa that colonize the seafloor, are they actual physical shape of the organisms, or changes in seafloor morphology as stated in lines 137-138?

line 41: what are the 'implications'. Is there space in the abstract to make a brief reference to the one or two most important points that need to be made?

line 555: Please define the acronym 'EOM'.

Figures 1 and 6 and their captions: I have already made this request in an earlier review. Please make sure that not only the figures, but the names and numerical identification of sites in the captions, and the chart in 6b are all consistent. For example, the Hakon Mosby mud volcano is labeled '5' in Figure 1, whereas the Hakon Mosby mud volcano is '7' in Figure 6b. The names of sites '8' and '9' are not in the caption for Figure 1.

Lines 653 and 667: Please use 'because' rather than 'as' for clarity.

Line 663: Please delete '(s) and vents (v)' because these abbreviations are not used.

Table 2 caption: 'n' and 'l' are not explained. Please clarify the meaning of these abbreviations used in the table.

Supplementary Figure 1 caption: Please delete the sentence 'The coordinates...decimal degrees.' because they are illegible on the photo.

Regarding the authors response to Reviewer 3.

1. Gas hydrate structure - They did remove the discussion of gas hydrate structure and classification, but Ruppel and Kessler (2016) was not added to the reference list as claimed by the authors. I think it is not necessary to add it.
2. Gas flare - The authors did address the comment about the inconsistency between the ROV video and the gas flare that disappears near the ocean surface. They improved Figure 2, using suggestions from Reviewer 3. However, I think reviewer 3 is asking for something that cannot be resolved with the available data. The authors attempted to provide one possible explanation for the persistence of an apparent gas flare – oil and/or a hydrate skin surrounding the bubble, which is reasonable. The gas flare is remarkable and consistent with being driven by a petroleum system at depth, which is supported by the molecular and isotopic data for the gas and extractable organic matter in the oil-soaked sample. What the authors do not mention is that the ROV video site and the seafloor source for the gas flare could be proximal, but spatially separated enough that the ROV scanning sonar would not have picked up gas bubbles emanating from the seafloor. I understand that there is a volumetric argument in that the bubbling observed by a ROV doesn't match the apparent magnitude of the apparent gas bubble plume that Reviewer 3 is posing, but this is not resolvable with the data available.
3. The color question – the authors removed discussion of hydrate color and speculation about its significance, except for what was clearly observed.
4. Warming bottom water – The authors have addressed Reviewer 3 comments by acknowledging the complexity of assessing the impact of warming bottom water on the stability of gas hydrate exposed on the seafloor.
5. Spongy texture – It is not clear to me what Reviewer 3 was seeing in the photo. I suggest to the authors to use the word 'vesiculated' rather than 'spongy' to describe the texture. It is not unusual for small gas bubbles to be incorporated in chunks of gas hydrate, which would cause the vesiculation – comparable to a vesiculated basalt – filled with gas vesicles. I think Reviewer 3 misinterpreted the spongy texture description as one of biological significance.

Reviewer #4

(Remarks to the Author)

I want thank the author for taking all the suggestion into consideration. In my opinion the manuscript is ready for acceptance.

made.

REVIEWER COMMENTS

Reviewer #1 (Remarks to the Author):

- Although the discovery of the deepest yet discovered chemosynthetic community is rather surprising and of interest to the scientific community, the manuscript has significant deficiencies that render it unsuitable for publication in Nature Communications. For example, **the caption for Figure 4 does not match the figure, instead it is for Figure 5.**

We appreciate the reviewer's comment. We have now corrected the caption of Figure 4.

Of most concern is the sediment coring and sampling methods. **Gas exsolution and gas hydrate decomposition were not mentioned and it is surprising the authors did not observe rapid degassing and sediment extrusion once the cores arrived on deck.**

We appreciate the reviewer's comment, which has helped us improve the clarity of our manuscript and, specifically, the sampling description. In our initial submission, we may not have adequately described the modalities of gas hydrate decomposition during the retrieval of the blade core, as concisely mentioned in lines 174-175. In the chapter "Mound morphology and composition", we now mention that the decomposition of gas hydrate has already started during the ascent of the ROV (because of the change in temperature and pressure) as visible in ROV live streaming, and as soon as the blade corer was brought on board. From the gas hydrate collected using the blade corer, we observed gas exsolution, also evidenced by igniting the methane released from a small piece of decomposing gas hydrates using a lighter. Because we were aware of the challenges associated with gas hydrate decomposition, we took immediate action once the blade corer was on deck and moved the blade corer to a cold room maintained at around 4°C to decelerate the decomposition and preserve the sample integrity. We have now provided the image from the seafloor showing the ROV arm collecting the gas hydrate from one of the Freya gas hydrate mounds (Supplementary Figure 1a) and also the image of the blade corer on deck filled with gas hydrate and sediment (Supplementary Figure 1b). We have also removed the information related to the gravity corer, since, it did not add anything to the paper as also indicated by Reviewer #2.

The anomalously high **85.9 mM** concentration of methane in a surface seafloor sample does not make any sense, unless the sample was collected using a pressure core barrel.

We appreciate the reviewer's observation regarding the anomalously high methane concentration of 85.9 mM in the surface seafloor sample. The same observation was also made by Reviewer #3. Upon further examination, we agree that this elevated concentration was probably an outlier, likely influenced by the presence of gas hydrate fragments within the sample and leading to an aberrant value. We removed this value from the current version of the manuscript. We acknowledge that accurate methane concentration measurements in hydrate-bearing sediments typically require specialized sampling techniques, such as pressure core barrels, to prevent hydrate dissociation and methane loss. However, we did not have such an instrument on board.

Thus, the gas analyses are not representative of the in situ concentrations as plotted in Figure 5 and it is not clear if the units of methane concentration are for the sediment pore fluid.

We apologize for the error in the caption of Figure 4, as noted by the reviewer in the first point. We have corrected the caption, and Figure 4 now accurately presents the molecular and isotopic composition of the gas hydrate sample contained in the blade corer (4 replicate subsamples were obtained from the same hydrate sample that can be seen in Supplementary Figure 1a and b. We removed the gas concentration profiles of the gravity core that were previously presented in Fig. 5 as they are not relevant to this manuscript, and it was also suggested by Reviewer #2.

The discussion of gas hydrate structures in lines 275-286 is not relevant to the manuscript and is speculative.

We would like to clarify that the section in question (lines 275-286) is based on references from already published papers. Our intention was to provide a concise explanation of the different structures of gas hydrates, which is pertinent to the context of our study. This background information helps understand the type of gas hydrate we are investigating. However, we acknowledge the reviewer's concern about this part of the text, which has also been commented on by Reviewer #3, explicitly referring to the type of gas hydrate. In response, we calculated the gas hydrate stability limit using in-situ conditions and the gas composition measured in our hydrate sample. The modelling results (added in the methods chapter), along with the measured gas released by hydrates, shows that the structure is type sII. This has been now described in the main text.

There is a substantial amount of speculation and inference throughout the text which is not supported by the data collected.

For example, lines 281-282, 305-312, and 326-331.

lines 281-282..... *The Freya mounds comprise gas hydrate Structure II because of the measured heavier thermogenic hydrocarbons.*

As explained above, we performed a calculation that allowed us to prove that Freya mounds comprise gas hydrate Structure II (sII). In fact, we calculated the gas hydrate stability limit using in-situ conditions and the gas composition measured in our hydrate sample. Structure I and H, although theoretically stable, are not compatible with the measured gas composition so they

cannot represent the dominant mass of the deposits. This has been now described in the main text.

Lines 305-312..... *The oil composition in gas*

*306 hydrate samples and sediments from the Molløy Ridge is consistent with source layers
307 of the Tertiary period or younger (<65 Ma) that reached the mid-oil-window stage. The
308 oil potentially derived from the Arctic Azolla blooms during the Eocene epoch or
309 organic-rich sediments deposited during the Miocene, as also proposed for PKF oil
310 seeps [36]. The similar composition of this oil suggests that fluids migrating at PKF at
311 80-100 m on the Svalbard margin and the Freya mounds at 3640 m water depth, more
312 than 100 km distant on the Molløy Ridge, are fueled from the same reservoir.*

We acknowledge the reviewer's comments, and modified the main text. We based our observations on the results obtained from geochemical analyses, which are carefully described in the methods chapter, using methods well accepted in a well-equipped and ISO-certified lab. We have explained that the analyses we performed on the oil from hydrate deposits and sediment allowed us to conclude that they contained oil from a Miocene or younger source. This conclusion is based on oil biomarkers and in particular on the abundance of compounds consistent with high angiosperm inputs, with only traces of gymnosperm. In the current version of the manuscript, we now clearly report this in the chapter "Mound morphology and composition." In response to the reviewer's concerns, we have further clarified our analyses and simplified the text to improve clarity and comprehension. We hope these revisions address any misunderstandings and reinforce the scientific basis of our conclusions.

Lines 326-331 *Additionally, we observed other bubbles that maintained a conventional rounded*

*327 appearance and had a dark colouration in the ROV video, suggesting a different
328 composition. Those spherical dark bubbles rose more slowly than the methane
329 bubbles, and we hypothesise that they are filled with oil, as observed at the shallower
330 PKF oil seep sites [36]. A scoop sample collected for fauna analysis contained oil and
331 emitted a strong hydrocarbon odor.*

To address the reviewer's concerns, we have included a video in the supplementary material that demonstrates the presence of methane bubbles at the Freya gas hydrate mounds. However, due to the unavailability of a video with suitable resolution, we were unable to provide clear evidence of the oily bubbles. Consequently, we have decided to remove the observation regarding the oily bubbles from our discussion.

The isotope and C1/C2+C3 data can be summarized in a sentence or two to establish the general source of the gas, but going beyond that and also including a 4 panel Figure 4 is not relevant to the manuscript.

We thank the reviewer for the suggestion. We have revised the geochemical section and removed Figure 5, which presented irrelevant gas data from the gravity core. Figure 4 now includes the molecular and isotopic composition of the hydrate in relation to genetic fields and is, therefore, essential for the manuscript.

Reviewer #2 (Remarks to the Author):

Review of:

Unveiling seafloor gas hydrate mounds with chemosynthetic fauna at 3640 m deep in the Mollø
Ridge, Greenland Sea
by Warren Wood

Recommendation: publish with minor revision

This manuscript describes data acquired around a methane hydrate mound in unprecedented water depths, and analysis showing that fauna whose taxa favor those found at a hydrothermal vent. This is truly a remarkable find - the deepest hydrate mound (3640 m deep) with the tallest known methane plume in the water column. Also remarkable is the observation that fauna with similar taxa are being nourished by significantly different sources. These findings should be published, and Nature Communications is the appropriate publication.

This manuscript contains significant qualitative description, which in my opinion is warranted given the unusual depth of this hydrate mound complex and the similarity in associated fauna to hydrothermal vents.

However, I would prefer a sharper conclusion. Figure 7 is the ideal place to make this point, but the data displayed here could be far better focused. At least a box around the Freya and Jotul site data (with labeling to indicate cold seep vs. hydrothermal vent) highlighting the faunal similarity between two vastly different types of seepage. It's OK to be a little redundant when conveying the point of the manuscript.

We acknowledge the reviewer's comment and changed the figure following the suggestions.

I would also prefer a discussion that is more focused on supporting support the main conclusion. The authors have an important point to make, but it is not too complicated, and this is not a "long-format" article. The discussion surrounding the main point (descriptions of other sites, and details of the fauna) are necessary, but their purpose is to provide context that should support the main point. **To that end it is not clear that the manuscript requires figures 4 and 5 perhaps one would suffice.**

As suggested by the reviewer, we refined the discussion to ensure that the descriptions of other sites and details of the fauna are more directly tied to our main point, providing necessary context without detracting from the core message. As previously mentioned, we have removed Figure 5 and we have now included the information from Figure 5 in the new Figure 4.

Minor comments:

Figure 3 caption – It might be over interpreting the image to call it a gas blow out. Unless there is supporting evidence, it could just as well be a manifestation of a slow

dissolution event.

We followed the reviewer's and now indicate in the caption that the depressions observed at Freya can result from gas blowout or slow dissolution events.

Lines 95-97, regarding depths of seeps and vents

This seems inconsistent with the previous text describing hydrothermal venting in the 500-750m depth range, and cold seeps at 70-800 m water depth. Or are the cold seeps at similar depths significantly far away, or oceanographically removed?

We have revised the text in that paragraph to clarify that the difference in depth between vents and seeps applies to all the sites north of 73°N latitude (and the shallower vent fields of Soria Moria and Troll Wall at 71°N are largely inhabited by fauna known from non-chemosynthetic habitats).

Lines 351-356

I do not see the explanation for how the shapes and morphologies of the hydrate mounds indicates their stage of evolution, or more specifically, the stage of population evolution the authors state. It seems that the nutrients that sustain the fauna might be affected by sediment cover, but not necessarily the shape of the mound.

The dissociation of hydrate (releasing gas and freshwater, and eroding the substratum) represents a physical disturbance for sessile fauna that have colonised the surfaces of the hydrate mounds. Availability of methane may also be reduced locally once most of the hydrate has dissociated from a structure. The pit-like depressions form where mounds have collapsed as a result of hydrate dissociation. They have patches of depauperate fauna, dominated by Stauromedusae and motile species, in contrast to the *Sclerolinum* forest and maldanid tubeworms occupying the mounds of intact hydrates. This change in faunal composition between earlier and late-stage "mound" morphologies is therefore consistent with physical disturbance from hydrate dissociation and/or subsequent local waning in methane supply. We have now clarified the text of this paragraph.

Figure 7 – might be helpful to repeat the color and symbol legend, to better distinguish between the cold seeps and hydrothermal vents.

We have revised the Figure 6 (previous Figure 7) as the reviewer suggested.

Lines 399-400 Could modern (last few hundred years) ocean currents be responsible for similarities along depths instead of across depths?

The mid-depth and deep waters of the region are dominated by the inflow of North Atlantic waters through the Fram Strait (Jones, 2008; Circulation in the Arctic Ocean, Polar Research 20: 139-146). Contemporary hydrography, therefore, does not correspond with the depth-driven similarity of fauna at chemosynthetic habitats in the region.

Note to the editor as well as authors: Lines 424-434 describe the political/environmental/legal impacts of the scientific results, and a context within which

they are viewed. I have no expertise to comment intelligently on “international obligations”, or if it should even be discussed in this manuscript.

Noted. Our experience indicates that a wider context of results (where they may potentially inform the development of environmental management plans) is common for research in this field, e.g. in the final paragraphs of similar papers (such as <https://www.nature.com/articles/srep39158>).

Reviewer #3 (Remarks to the Author):

Panieri et al. present results describing a newly discovered gas hydrate field along the Molløy Ridge at 3640m. I commend the authors on their discovery; however, the paper needs a major revision before it can be considered for publication in Nature Communications. In the following pages, I present comments/questions by line number. I hope that my comments help the authors improve the manuscript.

Detailed comments for the Authors

Line 28: Here and elsewhere in the manuscript: Gas hydrates do not fuel methane seepage. Quite the contrary, methane seepage, when occurring within an appropriate T-P window, fuel methane hydrate formation. This sentence should be re-written along the lines of "Seafloor seepage of methane can lead to the formation of methane hydrate. Methane seepage also supports chemosynthetic communities in the deep sea."
We have edited the manuscript to make the distinction.

Line 30: Suggest changing to: "...vents found at greater depths along and near active spreading centers". Note: Hydrothermal fluid discharge is not limited solely to mid-ocean ridges. Substantial off-axis fluid discharge occurs in both the Atlantic (e.g., Lost City) and in the Pacific (e.g., YBW-Sentry).
We have included the sentence as suggested (if the Abstract word-limit allows).

Line 43: It is worth including oil production activities in the last sentence "...deep sea mining and oil production activities in the region"
We acknowledged the reviewer's comment and changed the text accordingly.

Line 52: In the Arctic, structure II gas hydrates are actually stable at much shallower depths at ultracold temperatures (-2°C; see the work in the Kara and Laptev Seas).
We acknowledged the reviewer's comment. After further reading of recently published papers, we have decided to maintain the 300 m water depth limit in the text. Various papers, including those focused on the Kara and Laptev Sea, do not specifically address gas hydrates at shallower depths, but we are open to including a specific paper if the reviewer can suggest one.

Line 59: The Milkov 2004 reference for the size of the gas hydrate reservoir is out of date.

A more recent estimate is available in: Lee, M.W., B.J Phrampus, A. Skarke, and W.T. Wood, 2022. Global estimates of biogenic methane production in marine sediments using machine learning and deterministic modeling. *Global Biogeochemical Cycles*, <https://doi.org/10.1029/2021GB007248>. Though this paper focuses on biogenic methane production, they also estimate global stores of gas hydrate -- "Our estimated range, as indicated by error bars, of carbon sequestered in hydrate is 62–6,951 Gt." Given the robust modeling approached applied by Lee et al., this paper is worth mentioning.

We followed the reviewer's suggestion to change the reference to the more recent paper (Lee et al. 2022).

Line 62: Biogenic methane has also been documented in the Gulf of Mexico, see Klapp et al. 2010, *Marine Petroleum Geology*, <https://doi.org/10.1016/j.marpetgeo.2009.03.004>.

We followed the reviewer's comment and added the suggested references.

Line 64: This sentence is misleading and confusing to the non-expert. Here again, gas hydrate is defined as a type of cold seep. That is wrong: A methane cold seep can exist without gas hydrate, BUT gas hydrate cannot exist without a source of methane via methane seepage. It is the oxidation of methane and the coupling of AOM to sulfate reduction, that produces the sulfide that fuels many of the chemo-symbioses found at methane seeps. However, the sulfide is not only from coupling to AOM, oxidation of dissolved organic matter and higher alkanes also fuels sulfate reduction and sulfide production. Please make this clear.

We acknowledged the reviewer's comment and changed the text to avoid any confusion in the definition of a cold seep.

Line 106: This sentence is poorly constructed. You are trying to make too many points and the reader will be confused: Arctic blooms...later depositions of organic-rich sediments during the Miocene potentially created seabed hydrocarbon reservoirs...". First, organic-sediments do not create hydrocarbon reservoirs. Second, is it relevant that the sediments contain freshwater (Azolla ferns) and other materials? Please clarify and focus this sentence.

We now clarified the scientific base for our source rock correlation between Molløy and Prins Karl Forland. This young source rock will be further investigated in this region and as we say in the new version of the manuscript, we draw a first line of comparison. We believe this potential link between the source rock of the deep and shallower adjacent seeps is worth mentioning.

99 *The Molløy Ridge is a slow to ultraslow seafloor spreading centre in the Fram Strait,*
100 *trending north for ~60 km from the Molløy Fracture Zone at ~79.1 °N to the Spitsbergen*
101 *Fracture Zone at ~79.7 °N [20]. The seafloor depth of the ridge axis varies from ~5000*
102 *m at its southern end, rising to ~102 1500 m on an Oceanic Core Complex midway along*
103 *the ridge, and descending to ~4000 m at the northern end [20]. The formation of the*
104 *Molløy Ridge began after the Norwegian-Greenland Sea opened at ~56 Mya [21], and*
105 *most likely the seafloor spreading at the current Molløy Ridge started at ~20 Mya [22].*

106 Arctic blooms of the freshwater fern *Azolla* at 56 Mya [23] and later depositions of
107 organic-rich sediments during the Miocene potentially created seabed hydrocarbon
108 reservoirs in the region [24] [25], and serpentinisation on the flanks of the Molløy Ridge
109 may also provide an abiotic source of methane to seafloor sediments [26] [27].

We agree that this paragraph required some contextualization and focus so we moved the reference to the source rock further down in the text where we discuss the oil data.

Line 114: Seismic data cannot provide any information on the source of gas (biogenic vs. thermogenic); please clarify.

We thank the reviewer for the comment. The statement to which the reviewer is referring too is not our statement, but it refers to a statement from Chand, S., et al., Acoustic evidence of hydrocarbon release associated with the Spitsbergen Transform Fault, north of the Molloy Ridge, Fram Strait. *Frontiers in Earth Science*, 2024. 12. However, we removed the sentence from our text because we also believed this conclusion could not be made using seismic data.

Line 117: Simplify - The source of these pelagic plumes have not yet been identified.

We thank the reviewer for the comment and simplified the text.

Line 125: Hydrate does not support chemosynthetic fauna, dissolved gas (and likely DOM) seepage fueling sulfate reduction does.

We have edited the sentence to make the distinction.

Line 145: You are reporting contiguous gas flares that reach from ~3600m to ~300m. The amount of gas/methane discharge required fuel flares 3300m tall would be tremendous. Does the ROV imaging of gas discharge at the seafloor support this. Rather than showing 2D images, I would like to see Qimera or Fledermaus 3D renditions of the plumes. Perhaps these hydrates sit atop a high-flow fault but more typically, deep (high pressure) seeps trap methane efficiently in hydrate. There is always some leakage around the hydrate, but for there to be sufficient discharge to fuel the enormous plumes you report, something unusual has to be going on. Do you have an explanation?

We acknowledge the extraordinary nature of the contiguous gas flares extending from approximately 3600m to 300m, and propose that this is due to a combination of hydrate and oil-coating of the bubbles, both slowing down the bubble dissolution and allowing the bubbles to travel longer distances through the water column. We added a short text describing the process. Regarding visualization, we used Qimera and Fledermaus software to provide 3D representations of the plumes along with the bathymetry, which we agree offer a more comprehensive visualisation. We modified Figure 2 to include this representation, and added a corresponding description in the method section.

We did observe methane bubbles from the area of the Freya mounds with the ROV, and we added a short video as Supplementary Data 1. Still, we did not follow the bubble train ascending with the ROV, as we did for other sites we investigated in much shallower water, like Prins Karl Foreland. However, we agree that the scale of these flares suggests an intriguing geological condition. In the paper from Chand et al., 2024, they suggest that the migration of gas occurs through boundary faults of the deep sediment-filled Spitsbergen Transform Fault depression.

We hope that our findings will encourage additional research in this area, potentially involving more advanced imaging techniques and access to icebreakers, which would greatly facilitate the study of this site.

Line 155: It would be extremely unusual for "gas seepage rising from the mounds". First, this infers that the mounds are cohesive and the images show that they are not intact mounds. Rather they are horseshoe shaped or they have a cavern (e.g., Fig. 3). Intact hydrate mounds block and effectively reduce methane seepage, such that any methane escape occurs along the edges of the hydrate structures. Please clarify.

We thank the reviewer for the comment and changed the text as suggested

Line 173: I am curious, why is the hydrate greenish-yellow? I have seen structure 1 hydrate with a blue-green tinge and I have seen yellow grading to orange hydrate (oil inclusion) but I do not recall ever seeing or reading a report of greenish-yellow hydrate?

We thank the reviewer for the observation regarding the colour of the hydrate described in our manuscript. The greenish-yellow hue observed in gas hydrate samples can be attributed to several factors, for example, impurities, like the presence of certain trace elements that can affect the colour of hydrates. For instance, iron compounds give a yellow or greenish colour or oil inclusions/pockets. In our samples, the presence of small amounts of oil we measured might contribute to the observed greenish-yellow hue. The only other report we found was here: <https://geoexpro.com/gas-hydrates-part-i-burning-ice/> in a text written by Lasse Amundsen and Martin Landrø, published Date: June 12, 2012.

Line 178: The alkane distribution in the sampled hydrate is quite typical for structure II hydrate. It is not "exceptionally" wet. The alkane composition is similar for other sites (for example the Gulf of Mexico), where structure II hydrate abounds.

We thank the reviewer for the useful suggestions that we implemented in the new version of the manuscript. The text now includes hydrate modelling, and more literature support.

Line 189: In the text , you report a methane concentration of 85.9 mM in surface sediments. It is simply not possible to measure such a high concentration unless special (a pressure corer) instruments are use. Else, the gases come out of solution during recovery. Furthermore, this high concentration - 85.9 mM - is not shown on Figure 5. Please clean up / correct.

We removed this sample from the manuscript as it was influenced by pieces of hydrates, which led to an anomalously high concentration value.

Line 223: The black precipitate is likely iron sulfide. Did you check to see if it was acid volatile?

We thank the reviewer for the question. We agree that the black precipitate was mostly likely iron sulfide. During this expedition, we did not check if it was acid-volatile, but we appreciate the suggestion and will consider this aspect in future studies to provide a more comprehensive characterization of the precipitate.

Line 271: There is certainly a potential for gas hydrate destabilization in the future, especially as the ocean warms. But, do you really expect the Arctic to warm at >3000m? I find that hard to believe. To make such a statement, you need to include evidence from physical oceanographic models showing deepwater - extremely deepwaters - are likely to warm. Else this is simply speculation and given the importance of this topic, the scientific community needs to be standing on firm ground and the way this is presented it is not.

We thank the reviewer for highlighting this critical aspect in the manuscript. We agree entirely with the reviewer's comment and do not expect significant warming at depths greater than 3000 meters in the Arctic. We intended to refer primarily to the potential impacts on biological systems of hydrate evolution (formation and dissociation). We acknowledge the importance of grounding such and have revised the manuscript to clarify this point. However, in Karam, S., Heuzé, C., Hoppmann, M., and de Steur, L.: Continued warming of deep waters in the Fram Strait, *Ocean Sci.*, 20, 917–930, <https://doi.org/10.5194/os-20-917-2024>, 2024, it is indicated that “in the Fram Strait, which is influenced by both Greenland Sea Deep Water and Eurasian Basin Deep Water, it is a bit more complex. Temperatures vary between -1.20 and -0.95 °C in the 1980s and, overall, warm to ~ -0.85 °C by the 2020s.” We added this observation in the text since it cannot be comp, tell excluded that gas hydrate at the seafloor (even if very deep) might be affected by warming.

Line 287: What is "Type 4"? This is an older paper and we have learned much since it was published. Hydrates often form along faults beneath the sediment. As gas seepage continues, hydrates are literally pushed upwards. Hydrates are buoyant and the sediment drape can function as an anchor. As hydrates move above the sediment-water interface, they begin to sublime, releasing dissolved methane (not so much bubbles - bubbles are emitted along the edges). As they age, they dissolve and caverns can develop, as can carbonate castes. Again it's the methane seepage - not the hydrate - that drives the community. Without gas seepage, the hydrate, and the chemosynthetic community, would not exist.

We sincerely thank the reviewer for their insightful comments. Regarding the question of clarifying what “Type 4” gas hydrate is, so this is a type of gas hydrate as defined by You, K., Flemings, P. B., Malinverno, A., Collett, T. S., & Darnell, K. (2019). Mechanisms of methane hydrate formation in geological systems. *Reviews of Geophysics*, 57, 1146–1196. . Specifically, Type 4 refers to “Concentrated hydrate at vent sites”. We agree that how we have written the chapter “Composition and Dynamics of the Freya Gas Hydrate Mounds” was a bit unclear and also caused comments from Reviewer 1, and this is why we have simplified this section to focus solely on the structure of the gas hydrate where “structure” refers to the molecular arrangement of the gas molecules within the hydrate. This is the most common way to classify the gas hydrate so we have removed the classification based on type, as the reviewer pointed out that the paper we referenced is somewhat outdated.

We also thank the reviewer for the detailed explanation of the dynamics between gas seepage, hydrates, and the surrounding environment. The point about the role of methane seepage in

sustaining the chemosynthetic community is significant and aligns with our observations. We have ensured that the manuscript reflects this critical relationship more clearly.

This part of the discussion is extremely speculative and needs to be anchored in more data showing specific examples and references. Without that, the "spongy" description needs to go. ROV video should be included to confirm the statements about pelagic bubbles and their source. The authors have not shown enough data to tell a story about "hydrate evolution". What you have is a snapshot that shows frequent hydrate, and this hydrate, I presume, occurs at the surface expression of a fault. A geological map and/or subbottom data would strengthen this argument. Hydrate evolution in other systems, like the Gulf of Mexico, is well documented. Please cite that literature.

We sincerely thank the reviewer for their insightful comment and agree, as mentioned for the comment above, that the chapter "Composition and Dynamics of the Freya Gas Hydrate Mounds" was a bit unclear. We have revised all the sections that could seem to be speculative, supporting our discussion with more data. Specific examples and references have been added to support our claims, ensuring that our arguments are well-grounded in existing research. Based on the suggestion, we have removed the "spongy" description from the manuscript. We have included ROV video footage in the supplementary materials to confirm our statements regarding pelagic bubbles and their source. This visual evidence provides a clearer understanding of the phenomena we describe. In addition, we made a new Figure 2 adding the Qimera or Fledermaus 3D renditions of the plumes and including its georeferenced position on the bathymetry. Regarding the hydrating part, we have expanded our discussion on hydrate evolution by incorporating thermodynamic modelling. We have cited relevant literature on hydrate evolution in other systems, such as the Gulf of Mexico, as suggested, to provide context and comparison. These references help to situate our findings within the broader field of hydrate research. However, we did not incorporate sub-bottom data in this manuscript by referring to a previously published paper (Chand et al., 2024) reporting sub-bottom observations.

We thank the reviewers for their thorough reading of our manuscript and their constructive comments, and also the Editor for giving us the possibility to submit an appeal. Below we have copied each review in full (in black text) and highlighted (main) reviewer comments in **black bold** text. We provide our response to them in blue.

Thanks to these requested comments and suggestions, we feel the manuscript has improved considerably from the previous version submitted and hope that our proposed revision will meet the criteria for publication in ***Nature Communications***.

Sincerely,
Giuliana Panieri (on behalf of all authors)

REVIEWER COMMENTS

Reviewer #1 (Remarks to the Author):

The authors are reporting on a remarkable discovery of methane-fueled seep communities at an unexpected water depth of ~3640m. They have more than adequately addressed the reviewer's comments except for their discussion of the type of gas hydrate found at the seep site. They infer that the gas hydrate is a Structure II hydrate based on the gas chemistry (lines 292-302).

This is not relevant to the paper and I suggest removing this section. An inference would easily be misunderstood by readers that Structure II gas hydrate was found at this site, which is not the case. Laboratory confirmation of the gas hydrate structure is necessary before making a statement about structural type.

We thank the reviewer for the positive feedback and additional suggestion. We agree that the gas hydrate structure was only inferred and it has been removed from the text accordingly.

In lines 302-304 an estimate of the depth of gas hydrate stability in the sediment is provided. Please include what geothermal gradient was assumed in this model.

We thank the reviewer for the comment. The geothermal gradient value used for hydrate modelling, together with the other parameters, is reported in the Methods section.

The comprehensive faunal comparison provided in Table 2 would be greatly improved by implementing the following suggestions:

(1) arrange the columns in order of decreasing depth from left to right;

We appreciate the reviewer's suggestion to arrange the columns in order of decreasing depth from left to right. We have implemented this change accordingly in the revised manuscript to improve clarity and facilitate the interpretation of depth-related data.

(2) use larger and different symbols for vent vs. seep designations for each site - either shape or color;

We thank the reviewer for the valuable suggestion to differentiate vent and seep site designations using larger and distinct symbols, either by shape or color. We have implemented

this modification in the updated figures to enhance visual distinction and improve the clarity of site categorization.

(3) highlight the chemosynthetic-dependent taxa in the family column. Except for specialists, most readers will not easily recognize the families of these taxa.

We chose not to highlight the chemosynthetic-dependent taxa in the family column because doing so might inadvertently suggest that all members of that family are chemosynthetic-dependent, which is not the case. For example, while some species within the family Rissoidae are indeed associated with chemosynthetic environments, many others are not. To avoid conveying an overgeneralization and to ensure taxonomic accuracy, we opted to indicate chemosynthetic dependence at the species or genus level in the relevant columns, rather than at the broader family level.

I have attached my version of a reorganized Table 2 to this review.

We appreciate the reviewer's suggestion to reorganize the Excel table for improved clarity. We have restructured the table to more closely align with the reviewer's proposed format. Specifically, while we adopted the overall organization, we maintained the first row as the site name rather than depth. This choice was made to preserve immediate site identification, which we consider crucial for user navigation and data interpretation. Depth information remains clearly presented in subsequent rows/columns to ensure accessibility and coherence.

I am not sure if the depth and spatial arguments are strongly supported by the data presented in this table. The issues I see, which are inherent in a discovery like this and a comparison to other regionally relevant chemosynthetic communities is that they are under-sampled. Paucity of representative organisms may be due to numerous factors, such as limited and selective sampling, thus the collection of organisms is not representative. I think it is difficult to make convincing generalizations at this point. Please re-examine the discussion in light of what Table 2 shows, discuss its limitations, and modify the text as appropriate.

We have revised the Discussion to acknowledge the limitations of resolution in our samples and data, for example in the new penultimate paragraph. The nature of the samples and dataset that we have analysed is comparable to that used in published studies and interpretations of biogeographic patterns for chemosynthetic habitats in other regions and we have added the refernces of the relatove papers in the main text.

There are site labeling inconsistencies between Figures 1 and 6. The numbered sites do not correspond with each other.

We appreciate the reviewer's suggestion and have corrected the site labelling in both Figures 1 and 6 to ensure consistency. The numbered sites now correspond exactly between the two figures.

The Figure 1 legend does not describe sites 8 and 9 on the map. Sites names are not consistently located. Please label the maps in the same way and correctly name the sites in the figure captions.

We appreciate the reviewer's suggestion and have revised Figure 1 accordingly. The legend now includes descriptions for sites 8 and 9. We have also standardised the placement of site names and ensured consistent labelling across all maps. The figure captions have been updated to correctly name all sites.

In the caption for Figure 3, the word 'depressions' is used. I think 'collapse features' is a more accurate description. Please change 'depression' to 'collapse feature' throughout the paper.

We thank the reviewer for the suggestion. The term “depressions” has been replaced with “collapse features” throughout the manuscript, including the caption for Figure 3.

In addition, there is no evidence for a 'gas blowout'. This implies evidence has been observed for a significantly pressured gas pulse that created the morphologic feature. Please use a more descriptive term that does not imply process.

We thank the reviewer for the comment. As suggested, we have replaced the term “gas blowout” throughout the manuscript. We have chosen to use the term “collapse feature”, as the reviewer suggested in the previous point, to provide a neutral, descriptive designation that does not imply a specific formation process or the presence of significantly overpressured gas. We think that this terminology better reflects the morphological nature of the feature as we observed, without inferring any causative mechanisms for which we do not present direct evidence in this manuscript.

Line 643 - 'Borealis in [13]' does not make sense. Please correct.

As suggested, we have corrected the text by removing the word “in,” leaving only the name “Borealis Mud Volcano” along with the reference [13].

In Figure 2, the lines are not dashed as mentioned in the caption. Please correct the figure.

As suggested, we have corrected the figures substituting the original lines with a dashed line.

The statements in Lines 99-103 are awkward and it is difficult to understand the point(s) that are being made.

As suggested, we have corrected the text.

Line 284 - I think 'first order' would be clearer than 'first step'.

As suggested, we have changed the text.

Reviewer #3 (Remarks to the Author):

In this revision Panieri et al. have refined their results describing a newly discovered gas hydrate field along the Molløy Ridge at 3640m. Despite their extensive revision, some issues remain that must be resolved before the paper is considered for publication in Nature Communications.

Detailed comments for the Authors

For the distribution of structure II gas hydrates in the Arctic, please see -- Reviews of Geophysics. The interaction of climate change and methane hydrates, Carolyn Ruppel & John Kessler, 2016.

We are familiar with Ruppel & Kessler (2016) and have included it in the reference list. However, in line with the comments of other reviewers, we have followed the suggestion to remove the discussion classifying gas hydrates in order to maintain the manuscript's focus on supported observations.

I made this comment on the original paper and my concern remains. "You are reporting contiguous gas flares that reach from ~3600m to ~300m. The amount of gas/methane discharge required fuel flares 3300m tall would be tremendous. Does the ROV imaging of gas discharge at the seafloor support this. Rather than showing 2D images, I would like to see Qimera or Fledermaus 3D renditions of the plumes. Perhaps these hydrates sit atop a high-flow fault but more typically, deep (high pressure) seeps trap methane efficiently in hydrate. There is always some leakage around the hydrate, but for there to be sufficient discharge to fuel the enormous plumes you report, something unusual has to be going on. Do you have an explanation?"

The reviewers added a video, but the discharge shown in the video is not consistent with the description of the acoustic flare shown in Fig. 2. It is strange that no dimensions are given for the flare in Fig. 2. Based on my experience mapping methane seeps in the Atlantic and Pacific, the bubble discharge shown in the video is inconsistent with the type of discharge required to generate a plume that reaches thousands of meters above the seafloor. A more sophisticated analysis of the geophysical data is needed to generate a cohesive, convincing argument.

Our observation regarding the height of the flare is based on hydroacoustic data showing a clear signature of gas bubbles in the water column. This observation is consistent with previous work (reported in the main text cited the paper by Thorsnes et al. 2023), who documented flares in the same location reaching between 1770 m and 3355 m in height. We therefore see no reason to misinterpret the hydroacoustic signal. The new figure 2 is a Fledermaus 3D image of the plume, as suggested by the same reviewer in the first round of revision.

As discussed in the manuscript, we suggest that the flat shape of some rising bubbles, likely due to coatings of oil and/or gas hydrate, reduces their ascent velocity, increasing their

residence time in the water column and thereby their potential to travel greater vertical distances toward the surface.

It is widely recognized that hydroacoustic observations of bubble streams often appear far more prominent than their visual manifestation in video footage as it is indicated in the main text also citing previous papers. This explains why the supplementary video does not depict a visually "giant" flare, even though the hydroacoustic data clearly indicates an extensive vertical extent. In the Figure 2, the plume is shown alongside temperature and salinity profiles, highlighting how high the bubble stream is rising (up to ~290m depth as shown by the black line).

I made the following comment on the original manuscript "I am curious, why is the hydrate greenish-yellow? I have seen structure 1 hydrate with a blue-green tinge and I have seen yellow grading to orange hydrate (oil inclusion) but I do not recall ever seeing or reading a report of greenish-yellow hydrate?"

The authors provided this resource as evidence of "greenish hydrate":

<https://geoexpro.com/gas-hydrates-part-i-burning-ice/> in a text written by Lasse Amundsen and Martin Landrø, published Date: June 12, 2012. This reference shows a green-colored methane molecule inside a cage of yellow water molecules. The cartoon has nothing to do with the color of hydrate observed by the authors. The "green" color could be glauconite and the yellow could be organic sulfur. It is also possible/likely that yellow-colored Beggiatoa could be associated with hydrate. This point needs more clarification.

We agree with the reviewer that the yellowish color can also be related to bacterial encrustation, and we have incorporated this alternative explanation into the manuscript. The "greenish" hue we originally mentioned was based on direct visual observation on deck and was interpreted as potentially resulting from a thin petroleum coating on the hydrate surface. Such coatings can occur in seep environments where hydrocarbons co-migrate with methane, and oil films can adhere to hydrate surfaces, altering their apparent colour. However, as we lack high-quality photographic documentation to support this observation, we have removed the reference to the green coloration from the main text. We now describe only the predominant yellow hydrates, which are clearly documented in the figures.

In the original manuscript, I commented "There is certainly a potential for gas hydrate destabilization in the future, especially as the ocean warms. But, do you really expect the Arctic to warm at >3000m? I find that hard to believe. To make such a statement, you need to include evidence from physical oceanographic models showing deepwater - extremely deepwaters - are likely to warm. Else this is simply speculation and given the importance of this topic, the scientific community needs to be standing on firm ground and the way this is presented it is not." The author's response is unsatisfactory. Even if the bottom water warmed to +1°C (or more), hydrate would be stable at the extreme depths of this site given the pressure. Arguing that the hydrate will be unstable if deepwater warms to -0.85°C is baseless and shows a lack of understanding of hydrate dynamics in the natural environment.

We thank the reviewer for the constructive comment. We acknowledge that our original statement could have been interpreted as speculative and did not sufficiently reflect the thermodynamic constraints of hydrate stability at extreme depths. We fully recognise that, given the high pressures at >3000 m water depth, gas hydrates would remain stable even with substantial bottom-water warming. In line with the reviewer's recommendation, we have removed the sentence: "This warming trend could potentially affect the stability and distribution of gas hydrates in the region, adding another layer of complexity to their evaluation as an energy resource and their implications for climate change."

Regarding the last point about the hydrate being "spongy" - hydrate is never spongy. Solid hydrate is quite firm. Looking at Supp Fig 1, the "hydrate" appears like a combination of a sponge (actual biological organism!) and a Beggiatoa mat. The blade core sample is not convincing - the filaments are strange and not like any hydrate I have ever seen. I am left more confused than convinced by the additions to the paper.

We thank the reviewer for their detailed observations. We acknowledge that the use of the term "spongy" may have caused some confusion regarding the texture of the material; accordingly, we have revised the manuscript to use a more precise description (this change is underway).

However, we respectfully maintain that the structure shown in Supplementary Figure 1 is indeed gas hydrate, with the biological material observed on the surface identified as frenalate "Sclerolinum forest". Supplementary Figure 1a depicts the in situ hydrate sample together with its associated fauna, collected by the blade corer. Supplementary Figure 1b shows the sample obtained from the blade corer, in which frenalate Sclerolinum are still visible extending from the lower part of the core.

Reviewer #4 (Remarks to the Author):

The manuscript describes the discovery of the world's deepest known seafloor hydrate mounds and associated cold-seep fauna with taxa similar to hydrothermal vent in the area rather than to other, shallower cold seeps. The authors also report contiguous gas flares extending from approximately 3600m to 300m and the proposed mechanisms deserve publication on their own, as they demonstrate the huge contribution that ultra-deep methane systems can make to global methane cycling under specific conditions. These findings are of great interest for a vast community and should be in Nature Communications.

In reviewing the manuscript I also kept into consideration the previous round of revision, including the comment made by three reviewers and the work done by the authors to respond to the comments. Overall I believe that the authors addressed all the reviewers previous comments and that the manuscript is currently in great shape. I appreciate that the authors briefly discuss the wider context of results in term of environmental management plans. More should be done in this direction when findings can potentially impact ecosystem and resource management.

A few minor comments follow:

line 66. maybe worth mentioning that the chemosynthetic production is of prokaryotic origin since the rest of the paragraphs focuses on eukaryotes? Being Nat Comm a generalist journal the author should not assume that all the readers are familiar with chemosynthesis and the current paragraph might be misleading.

We appreciate the reviewer's suggestion and have clarified in line 66 that the chemosynthetic production is of prokaryotic origin.

Line 99-103. Some of the differences in the fauna between vents and cold seeps might also be driven by the different geochemical sources (and temperature regimes) supporting very diverse microbial communities in these ecosystems. Perhaps this could be mentioned citing relevant microbiological literature.

We have revised the text in the Introduction and Discussion to highlight differences in geochemical sources, and consequent differences in microbiology, as factors that may drive faunal differences, with additional referencing as suggested. In addition we have also noted that Jøtul vent fluids have high methane concentrations and thus biogeochemistry may drive similarities in the biota at the Jøtul Vent and Freya hydrate seeps.

Line 249-254. As s suggestion, an additional way to test for this is using a matrix of beta diversity between sites and testing its dependence on the pairwise distance between sites using a mantel test.

This is indeed an alternative way to test for the relationships that we explore, which produces the same outcome, and the tests that we present are equally valid (particularly given the sparse nature of the dataset here).

Line 351-361. I agree with a previous reviewer that the current study did not classify the morphological evolution of a site, rather describes a series of snapshot from diverse features in the site with the assumption (according to my best understanding) that they represent different time-points of morphological evolution. However, they are still distinct features observed at the same time point, and this can be further clarified also in line 134-1367.

We appreciate the reviewer's comment and clarified the text in both sections (lines 134–136 and 351–361) to explicitly state that the observed features are snapshots of diverse morphologies present at the same time, and their classification as stages of evolution is based on inferred processes rather than direct temporal evidence.

Line 406-420- Very interesting section of the discussion.

We appreciate the kind comment or the reviewer. Thank you!

We thank the reviewers for their thorough reading of our manuscript and their constructive comments, and also the Editor for giving us the possibility to submit the revised version. Below we have copied each review in full (in black text) and highlighted (main) reviewer comments in **black bold** text. We provide our response to them in blue.

Thanks to these requested comments and suggestions, we feel the manuscript has improved considerably from the previous version submitted and hope that our proposed revision will meet the criteria for publication in ***Nature Communications***.

Sincerely,

Giuliana Panieri (on behalf of all authors)

REVIEWER COMMENTS

Reviewer #1 (Remarks to the Author)

This manuscript has matured during the review process and I recommend publication after a few important corrections and clarifications are made.

We thank the reviewer for the positive feedback.

Abstract - lines 35-36: Please clarify by what is meant by 'display morphologies resulting from progressive stages of gas hydrate dissociation'. Are the morphologic changes, changes in the assemblage of taxa that colonize the seafloor, are they actual physical shape of the organisms, or changes in seafloor morphology as stated in lines 137-138?

We appreciate the reviewer's suggestion to clarify what we wanted to describe with the sentence 'display morphologies resulting from progressive stages of gas hydrate dissociation' that was due to a mistake in the grammar or way in which the sentence was written. We clarified in the Abstract that the "morphologies" refer to changes in seafloor morphology resulting from different stages of gas hydrate dissociation. The revised sentence now reads:

"The mounds display seafloor morphologies resulting from progressive stages of hydrate dissociation, which may also drive an ecological succession."

Line 41: What are the "implications"? Is there space in the abstract to briefly mention the one or two most important points?

We thank the reviewer for this constructive suggestion. We have revised the last sentence of the abstract to make the implications more explicit. Specifically, we now clarify that our findings contribute to understanding the connectivity between seep and vent ecosystems in the Arctic and their susceptibility to environmental and industrial disturbances. The revised sentence now reads:

“These findings have implications for understanding the link between thermogenic hydrocarbon systems and deep Arctic ecosystems, as well as their vulnerability to environmental change.”

line 555: Please define the acronym 'EOM'.

We thank the reviewer for the comment and defined EOM (Extractable Organic Matter).

Figures 1 and 6 and their captions: I have already made this request in an earlier review.

Please make sure that not only the figures, but the names and numerical identification of sites in the captions, and the chart in 6b are all consistent. For example, the Hakon Mosby mud volcano is labeled '5' in Figure 1, whereas the Hakon Mosby mud volcano is '7' in Figure 6b. The names of sites '8' and '9' are not in the caption for Figure 1.

We apologise for the confusion and thank the reviewer for pointing this out once again. All inconsistencies have now been corrected: Figures 1 and 6, their captions, and the numerical identification of the sites (including the chart in 6b) are now fully consistent.

Lines 653 and 667: Please use 'because' rather than 'as' for clarity.

The suggested changes have been made.

Line 663: Please delete '(s) and vents (v)' because these abbreviations are not used.

We left the abbreviations (s) and (v) as we have replaced (l) and (n) in the Table 2 with (s) and (v) to clarify.

Table 2 caption: 'n' and 'l' are not explained. Please clarify the meaning of these abbreviations used in the table.

As we replaced (n) and (l) in Table 2 with (s) and (v), the former don't need to be explained.

Supplementary Figure 1 caption: Please delete the sentence 'The coordinates...decimal degrees.' because they are illegible on the photo.

This sentence has been deleted from the caption of Supplementary Figure 1.

Regarding the authors response to Reviewer 3.

1. Gas hydrate structure - They did remove the discussion of gas hydrate structure and classification, but Ruppel and Kessler (2016) was not added to the reference list as claimed by the authors. I think it is not necessary to add it.

We confirm that Ruppel & Kessler (2016) has been removed from the reference list as it was ultimately not necessary.

2. Gas flare - The authors did address the comment about the inconsistency between the ROV video and the gas flare that disappears near the ocean surface. They improved Figure 2, using suggestions from Reviewer 3. However, I think reviewer 3 is asking for something that cannot be resolved with the available data. The authors attempted to provide one possible explanation for the persistence of an apparent gas flare – oil and/or a hydrate skin surrounding the bubble, which is reasonable. The gas flare is remarkable and consistent with being driven by a petroleum system at depth, which is supported by the molecular and isotopic data for the gas and extractable organic matter in the oil-soaked sample. What the authors do not mention is that the ROV video site and the seafloor source for the gas flare could be proximal, but spatially separated enough that the ROV scanning sonar would not have picked up gas bubbles emanating from the seafloor. I understand that there is a volumetric argument in that the bubbling observed by a ROV doesn't match the apparent magnitude of the apparent gas bubble plume that Reviewer 3 is posing, but this is not resolvable with the data available.

We appreciate the reviewer's summary and their understanding that certain aspects cannot be resolved with the available data.

3. The color question – the authors removed discussion of hydrate color and speculation about its significance, except for what was clearly observed.

Thank to the reviewer for acknowledging and appreciating our efforts.

4. Warming bottom water – The authors have addressed Reviewer 3 comments by acknowledging the complexity of assessing the impact of warming bottom water on the stability of gas hydrate exposed on the seafloor.

Thank to the reviewer for acknowledging and appreciating our efforts.

5. Spongy texture – It is not clear to me what Reviewer 3 was seeing in the photo. I suggest to the authors to use the word 'vesiculated' rather than 'spongy' to describe the texture. It is not unusual for small gas bubbles to be incorporated in chunks of gas hydrate, which would cause the vesiculation – comparable to a vesiculated basalt – filled with gas vesicles. I think Reviewer 3 misinterpreted the spongy texture description as one of biological significance.

We replaced "spongy" with "vesiculated" to describe the texture, as suggested by the reviewer.

Reviewer #4 (Remarks to the Author)

I want thank the author for taking all the suggestion into consideration. In my opinion the manuscript is ready for acceptance.

We thank Reviewer #4 for the positive evaluation and for recognising our efforts to address all prior suggestions. We are pleased that the manuscript is now considered ready for acceptance.

Table 2. Faunal families recorded papers (Vestri Arctic seeps (s) and vents (v)) filed from publisior other sites corinventories for Fri, this study, data bya and Jøtu from

Families	Aurora Vent Field (V)	Frey (s)	Jøtu (V)	Løkis Castle (V)	Håkon Mosby Mud Volcanic (s)	Vestnesa Ridge (s)	Storfordrenna+ Bjørnøyrenn (s)	Prins Karls Fjord (s)
Porifera	Cladorhizidae Fossilidae Theridae							
Cnidaria	Actinosolidae Bathyphelellidae Corallimorphidae Edwardsidae Hornathiidae Kadosaciniidae Luernariidae							
Arthropoda	Lysianassidae Mellitidae Munropsidae Nymphonidae Oedicerotidae Oregonidae Phoridae Phoxocephalidae Pleustidae Sebidae Stenothoidae							
Mollusca	Buccinidae Cocculinidae Cuspidaridae Hyalogyridae Lyonsteliidae Mytilidae Rissoidae Skeneidae Thyasiridae Xylorisculidae Yoldidae							
Echinodermata	Astropectinidae Bathyrchinidae Benthoplectinidae Elpididae Ophiopygidae Ophiuridae							
Chordata	Anarrichadidae Gadidae Rajidae Sebastidae Zoarcidae Microphallidae (platyhelminth)							

high & low
chemo & hydrothermal - deep data
taxa

3880m Aurora V
3640m Freya S
3020m Jøtu V
2350m Løkis Castle V
1250m Håkon V
1200m Vestnesa
350-390m Storfj.
350m Prins